# CaMKK2 and CHK1 phosphorylate human STN1 in response to replication stress to protect stalled forks from aberrant resection

Rishi Kumar Jaiswal[1], Kai-Hang Lei [2], Megan Chastain[3], Yuan Wang[4], Olga Shiva[3], Shan Li[5], Zhongsheng You[5], Peter Chi [2,6] & Weihang Chai [1] ✉

Keeping replication fork stable is essential for safeguarding genome integrity; hence, its protection is highly regulated. The CTC1-STN1-TEN1 (CST) complex protects stalled forks from aberrant MRE11-mediated nascent strand DNA degradation (NSD). However, the activation mechanism for CST at forks is unknown. Here, we report that STN1 is phosphorylated in its intrinsic disordered region. Loss of STN1 phosphorylation reduces the replication stress-induced STN1 localization to stalled forks, elevates NSD, increases MRE11 access to stalled forks, and decreases RAD51 localization at forks, leading to increased genome instability under perturbed DNA replication condition. STN1 is phosphorylated by both the ATR-CHK1 and the calcium-sensing kinase CaMKK2 in response to hydroxyurea/aphidicolin treatment or elevated cytosolic calcium concentration. Cancer-associated STN1 variants impair STN1 phosphorylation, conferring inability of fork protection. Collectively, our study uncovers that CaMKK2 and ATR-CHK1 target STN1 to enable its fork protective function, and suggests an important role of STN1 phosphorylation in cancer development.

Faithful DNA replication during each cell cycle is crucial to safeguarding genome integrity[1,2]. Replication forks face various obstacles such as DNA secondary structure, transcription-replication collisions, highly repetitive sequences, insufficient nucleotides, heterochromatic regions, DNA damage, and oncogene activation, which can block DNA polymerase progression and lead to replication stress. Replication stress is the major force driving the development of many diseases, including cancer, developmental defects, neurological diseases, and aging[3,4].

To maintain genomic stability, it is crucial to stabilize and restart stalled replication forks. During replication stalling, stretches of single-stranded DNA are formed at these forks and are bound by the replication protein A (RPA) complex. This triggers the activation of the ataxia telangiectasia and Rad3-related (ATR) kinase, which leads to the phosphorylation of its downstream effector checkpoint kinase CHK1.

CHK1 phosphorylates various targets and recruits them to stalled forks to facilitate fork stabilization and restart[5,6]. While ATR activation due to disturbed DNA replication is well-established, a recent discovery has unveiled a novel replication stress response pathway. Stalled replication generates cytosolic DNA, which activates the cGAS-STING pathway and results in the release of $Ca^{2+}$ from endoplasmic reticulum to the cytosol[7]. The elevated intracellular $Ca^{2+}$ concentration activates the calcium/calmodulin dependent protein kinase kinase 2 (CaMKK2), which in turn phosphorylates its downstream kinase AMPKα. AMPKα then phosphorylates Exonuclease 1 (EXO1), inhibiting its nuclease activity and preventing uncontrolled fork resection by EXO1[7,8]. Although many targets of ATR-CHK1 at stalled forks have been identified, EXO1 is the only known target of the CaMKK2-AMPK signaling pathway upon fork stalling.

[1]Department of Cancer Biology, Cardinal Bernardin Cancer Center, Loyola University Chicago Stritch School of Medicine, Maywood, IL, USA. [2]Institute of Biochemical Sciences, National Taiwan University, Taipei, Taiwan. [3]Office of Research, Washington State University, Spokane, WA, USA. [4]Department of Radiation Oncology, Rutgers Cancer Institute of New Jersey, New Brunswick, NJ, USA. [5]Department of Cell Biology and Physiology, Washington University School of Medicine, St. Louis, MO, USA. [6]Institute of Biological Chemistry, Academia Sinica, Taipei, Taiwan. ✉e-mail: vchai@luc.edu

Among all the mechanisms that protect and restart the stalled replication fork, replication fork reversal has emerged as a key mechanism. Electron microscopy analysis shows that a distinct DNA structure, commonly referred to as a "chicken foot" or reversed fork structure, forms after fork stalling through fork remodeling[9]. During fork reversal, the position of the fork moves in the backward direction away from the obstacle, forming a four-way DNA junction through the simultaneous annealing of nascent and template strands. However, the regressed arm formed during fork regression is vulnerable to nucleolytic attacks by various nucleases such as DNA2, MRE11, and EXO1. If reversed forks are not properly protected, unscheduled nucleolytic attack can result in nascent strand DNA degradation (NSD) and genome instability[10–12]. To protect stalled forks from nucleolytic attacks, numerous proteins have been reported to protect forks from aberrant nucleolytic degradation, including RAD51, RAD51 paralogs, PALB2, PARP1, BRCA1, BRCA2, BARD1, FANC porteins, BOD1L, WRNIP1, ABRO1, RECQ1, CtIP, SETD1A and many others[10,13–28].

Recently, we have found the human CTC1-STN1-TEN1 (CST) trimeric complex, consisting of CTC1, STN1, and TEN1, as a fork protector that antagonizes MRE11 from degrading nascent strand DNA[29]. The CST complex binds to single-stranded (ss) DNA and ss-ds DNA junctions[30,31]. Loss-of-function mutations in CTC1 and STN1 cause two complex genetic diseases Coats plus and Dyskeratosis congenita[32–38]. Moreover, CST has been implicated in tumor development, as STN1 variants are associated with various types of cancer[39–44], and STN1 deficiency promotes colorectal cancer development in young adult mice[45]. We have found that, in response to hydroxyurea (HU) treatment, CTC1 and STN1 proteins localize at stalled forks and inhibit MRE11-mediated NSD after fork reversal. Using purified CST proteins, we have found that CST can protect DNA from MRE11 degradation in vitro, and such protection is dependent on CST's ability to bind to DNA[29]. In addition, CST directly interacts with RAD51 under perturbed replication conditions and facilitates RAD51 recruitment to stalled forks[29,46,47]. CST depletion leads to ssDNA accumulation and chromosome fragmentation after HU treatment. These results suggest that CST may protect reversed forks from NSD in both RAD51-dependent and –independent manners[29,46,47].

The role of CST in maintaining genome stability extends beyond fork protection. CST facilitates dormant origin firing in response to replication stress, which aids in rescuing genome-wide replication fork stalling[48]. Additionally, CST is involved in the double-strand break (DSB) repair pathway by interacting with the shieldin complex and localizing to DNA damage sites in a shieldin- and 53BP1-dependent manner. It is thought that CST promotes non-homologous end joining, particularly when BRCA1 is deficient[49]. During normal DNA replication, CST disrupts CDT1 binding with the MCM complex, leading to decreased origin licensing[48]. Furthermore, CST promotes replisome assembly and origin firing in the S-phase of the cell cycle by increasing AND-1 and DNA Polymerase α (POLα) chromatin association[48]. At telomeres, CST facilitates efficient telomeric DNA replication, thereby preventing sudden loss of telomeres[50,51]. Additionally, CST coordinates G- and C-strand synthesis by directly interacting with telomere-binding proteins POT1-TPP1, which inhibits telomerase access to telomere ends[52,53]. The CST complex has also been studied in telomerase-negative ALT (alternative telomere lengthening) cells, where it regulates C-circle production by localizing at ALT-associated PML bodies[54].

Despite recent findings on the important role of CST in protecting the stability of stalled forks, the mechanism by which replication stress response pathways activate CST remains unknown. To shed light on this process, we sought to determine how CST is activated by fork stalling. Our previous work revealed that the STN1-OB domain contains an intrinsically disordered region (IDR) that is rich in polar residues, and a number of single mutations in this IDR causes genome instability and reduces RAD51 foci formation under replication stress[55]. In this study, we report that deletion of this IDR leads to NSD, suggesting that the STN1 IDR is vital for protecting stalled forks under perturbed replication conditions. We identified serine 96 (S96), located in this IDR, as a phosphorylation site that responds to replication stress. Disrupting S96 phosphorylation decreases STN1 localization at stalled forks and impairs RAD51 recruitment, leading to NSD and genome instability, while the phosphomimetic mutant protects fork stability and ensures STN1 and RAD51 localization at stalled forks. We have found that the ATR/CHK1 pathway phosphorylates S96 in response to replication stress induced by HU and aphidicolin (APH). We have also discovered that CaMKK2, a calcium-sensing kinase, phosphorylates S96 in response to HU treatment or the increased intracellular calcium concentration, suggesting that STN1 can be phosphorylated by CaMKK2 in a calcium-dependent manner. Our in vitro kinase assay, using purified CHK1, CaMKK2, and STN1 proteins, demonstrates that CHK1 and CaMKK2 are able to directly phosphorylate STN1 at the S96 position. Furthermore, we observe that two cancer-associated missense mutations at or near S96 diminish STN1 phosphorylation. Consistently, DNA fiber assay shows that these cancer-associated mutations are defective in protecting stalled forks from NSD. Interestingly, loss of S96 phosphorylation has no obvious effect on the CST complex formation and cellular localization, CST binding to DNA, CST interaction with POLα or RAD51. Although further investigation is needed to elucidate how the ATR/CHK1- and CaMKK2-mediated phosphorylation of STN1 regulates STN1 and RAD51 localization to stalled forks, our findings provide molecular insights into how cells respond to fork stalling to protect stalled forks from unscheduled nucleolytic degradation. These results reveal the importance of the STN1 IDR in maintaining fork stability under perturbed replication conditions, and highlight the role of the calcium-sensing CaMKK2 signaling pathway in activating fork protective proteins in response to replication stress.

## Results

### The STN1 IDR in the OB-fold domain protects stalled forks from NSD

CST and RPA are structurally similar as both contain multiple OB-fold domains. In the CST complex, CTC1 contains seven OB-fold domains, and STN1 and TEN1 each contains one OB-fold domain. In the RPA complex, RPA70 contains four, and RPA32 and RPA14 each contains one OB-fold domain[30,56,57]. Overlaying RPA32-OB/RPA14 with STN1-OB/TEN1 using PyMOL shows that the two structures superimpose upon each other (Fig. 1A). We have previously reported that despite the structural similarity between STN1-OB and RPA32-OB, STN1-OB contains a unique 26 aa IDR that is distinct from the IDR in RPA32-OB (Fig. 1A)[55]. This IDR connects β-strands 3 and 4, does not contain sufficient hydrophobic amino acids, and is mainly rich in polar and charged amino acid residues (Fig. 1A). Our previous study shows that this IDR is important for maintaining genome stability under replication stress[55]. Since CST protects the stalled replication fork from the MRE11 nuclease degradation under replication stress[29], we sought to determine whether this IDR played a role in fork protection using DNA fiber assays. We depleted endogenous STN1 using siRNA and expressed RNAi-resistant wild-type (WT) STN1 and IDR-deleted STN1 (ΔIDR) from U2OS cells, and then sequentially labeled replication tracks with chlorodeoxyuridine (CldU) and iododeoxyuridine (IdU) for 20 min, followed by 4 mM HU treatment for 3 h to induce fork stalling (Fig. 1B). Consistent with our previous report[29], STN1 depletion induced NSD, and RNAi-resistant WT-STN1 completely rescued NSD in STN1-depleted cells (Fig. 1B). Interestingly, STN1-ΔIDR failed to rescue the NSD caused by STN1 depletion (Fig. 1B). As RPA32 shows structural similarity to STN1, we then tested whether RPA32 overexpression could protect stalled forks in the absence of STN1. We found that RPA32 overexpression was unable to protect forks from NSD when STN1 was depleted (Fig. 1B), suggesting that the functions of

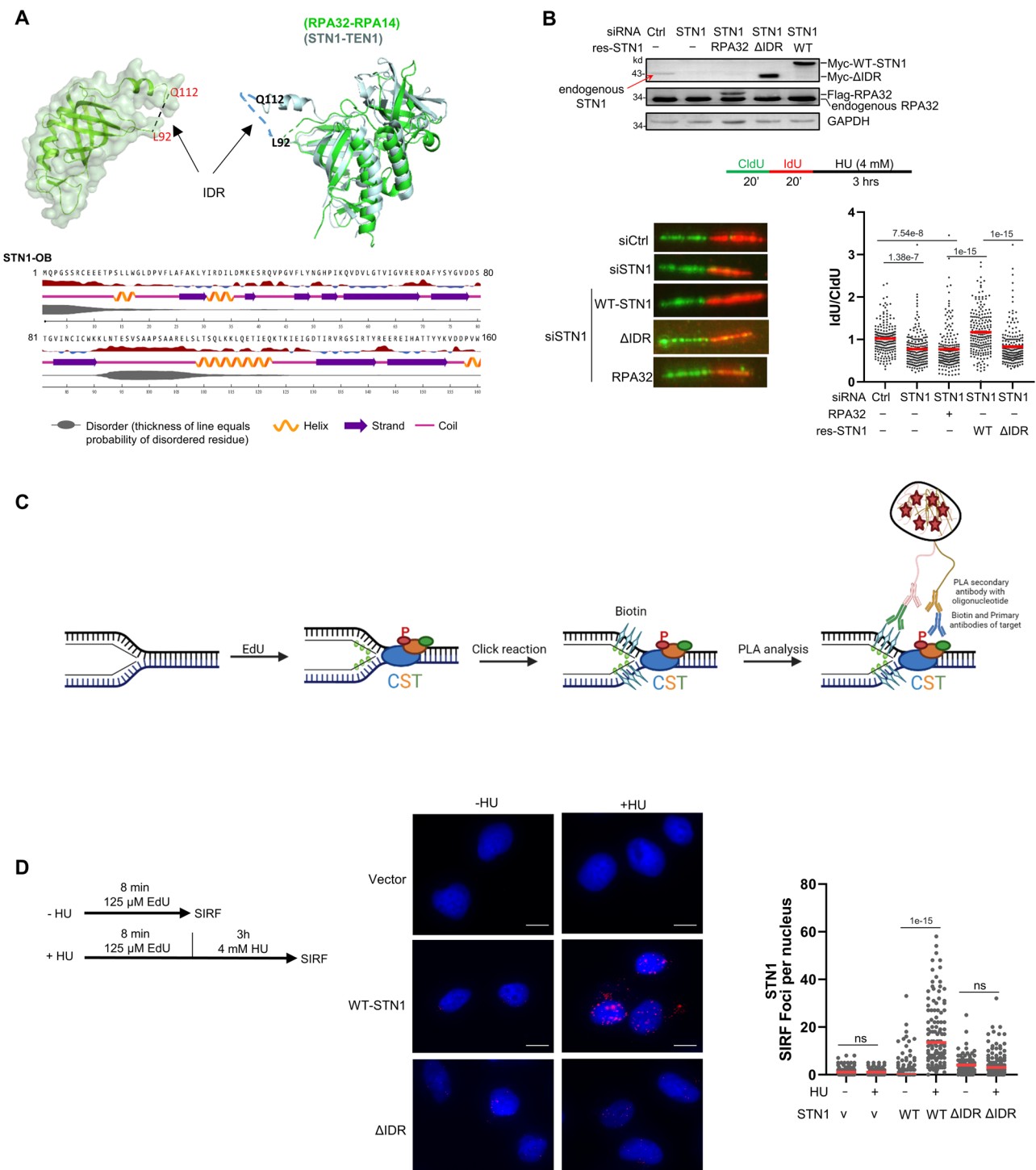

**Fig. 1 | The STN1 IDR in the OB-fold domain protects against NSD under replication stress. A** Top: Structure of the STN1-OB domain (PDB: 4JOI). PyMOL software shows that RPA32-OB/RPA14 and STN1-OB/TEN1 superimpose upon each other except the 26 aa (90–116) IDR present in STN1-OB. Bottom: NetSurfP-2.0 software predicts that STN1 90–116 is an IDR. The STN1-OB domain (aa 1–160) was used in prediction. **B** DNA fiber analysis of NSD in U2OS cells expressing RPA32, WT-STN1, or STN1-ΔIDR with concurrent knockdown of endogenous STN1. Flag-RPA32, Myc-WT-STN1, and Myc-STN1-ΔIDR were stably expressed by retroviral transduction. Three independent experiments were performed and the result from one experiment is shown. *P*: One-way ANOVA. The mean values are shown in red lines. *n* = 200 fibers were measured per sample in each experiment. Western blot shows STN1 knockdown and the expression of Flag-RPA32, Myc-WT-STN1, and Myc-STN1-ΔIDR. The expression level of endogenous STN1 is low (pointed by a red arrow). Note that Myc-STN1-ΔIDR migrates at almost the same position as the endogenous STN1. **C** Scheme of SIRF assay. Nascent strand DNA was pulse labeled with EdU to incorporate EdU at forks. Click chemistry was performed to covalently link biotin to EdU. Following incubation with primary antibodies (anti-biotin and anti-pS96) and secondary antibodies, PLA amplification was performed to visualize the proximity of phosphorylated STN1 to EdU-labeled forks. The scheme was created with BioRender.com. **D** SIRF detection of WT-STN1 and ΔIDR at normal and stalled forks. Myc-tagged WT-STN1 and ΔIDR were stably expressed in U2OS cells with retroviral transduction. Cells were pulse labeled with EdU for 8 min, then treated with or without HU (4 mM) for 3 h. Scale bars: 10 μm. Images with the red channel are provided in Supplementary Fig. 8. Two independent experiments were performed and the result from one experiment is shown. *P*: One-way ANOVA. Red line: mean. *n* = ~100 cells were measured per sample in each experiment. Source data are provided in the Source Data file.

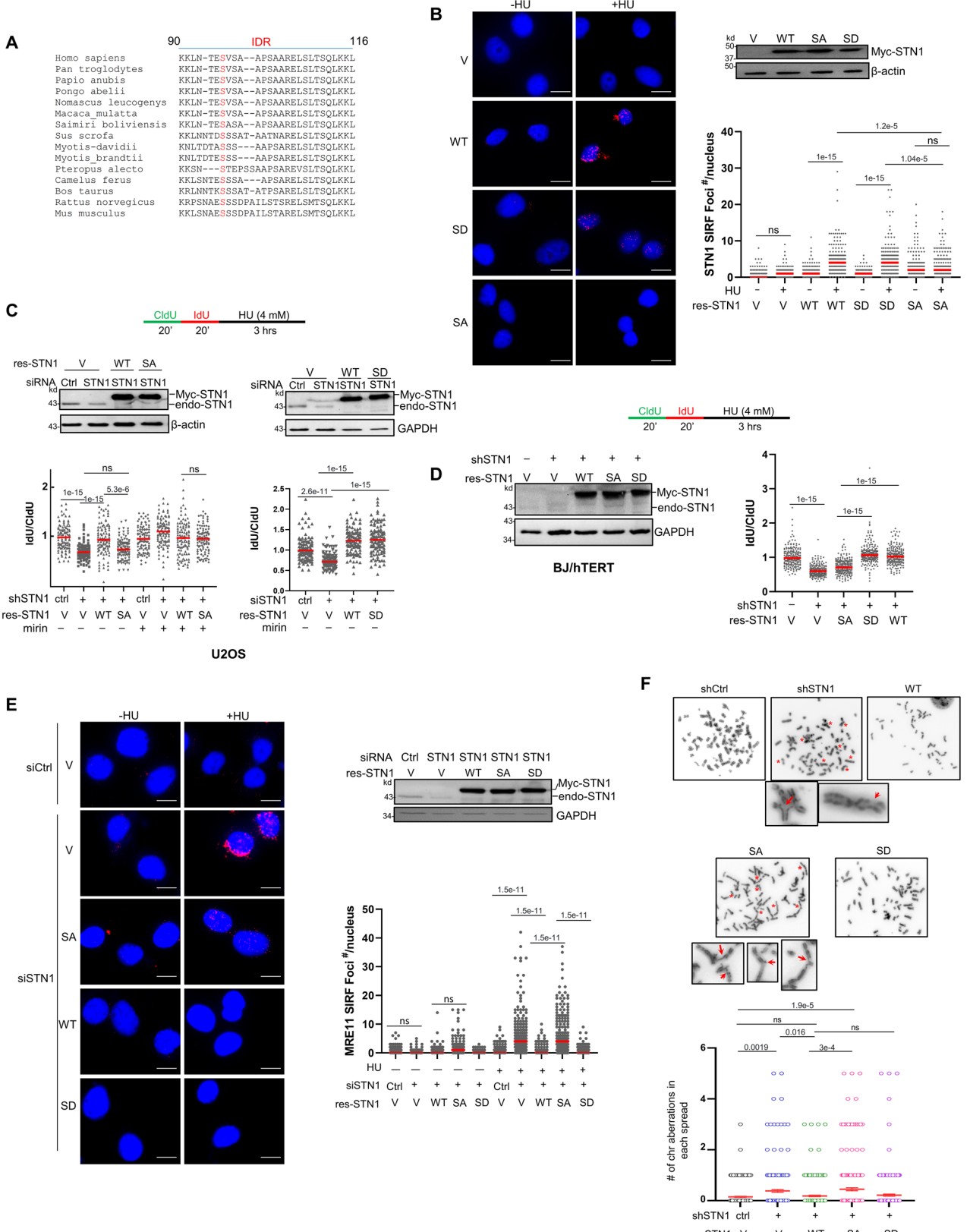

RPA32 and STN1 in protecting fork stability are not exchangeable. Consistently, non-denaturing BrdU staining showed that STN1 depletion caused an increase in ssDNA formation, which was rescued by the RNAi-resistant STN1 but not by RPA32 or STN1-ΔIDR (Supplementary Fig. 1).

Next, we determined whether the IDR was important for STN1 localization at stalled forks. The Myc-tagged WT-STN1 and ΔIDR were stably expressed with retroviral transduction in U2OS cells and the SIRF (in situ analysis of protein interactions at DNA replication forks) assay[58] was performed as published previously[29]. The SIRF assay allows

**Fig. 2 | S96 in STN1 IDR is essential for antagonizing MRE11-mediated degradation of nascent strand DNA and for protecting genome stability. A** Sequence alignment showing that S96 is conserved in higher eukaryotes. S96 is marked in red. **B** SIRF detection of S96A, S96D, and WT-STN1 at normal and stalled forks. Myc-tagged WT, S96A, S96D were stably expressed with retroviral transduction in U2OS cells treated with 4 mM HU for 3 h. Scale bars: 10 μm. Images with the red channel are provided in (Supplementary Fig. 9). Three independent experiments were performed and the result from one experiment is shown. *P*: One-way ANOVA. Red line: mean. *n* = 150 cells were analyzed per sample in each experiment. Western blot shows the expression of S96A, S96D, and WT-STN1. **C** DNA fiber analysis detecting NSD in the same U2OS cells described in (**B**). Three independent experiments were performed and the result from one experiment is shown. *P*: One-way ANOVA. Red line: mean. *n* = 100 fibers were measured per sample in each experiment. **D** DNA fiber analysis of NSD in BJ/hTERT cells co-expressing shSTN1 and RNAi-resistant S96A, S96D, and WT-STN1, which were expressed using retroviral transduction.

Two independent experiments were performed and the result from one experiment is shown. *P*: One-way ANOVA. Red line: mean. *n* = 200 fibers were measured per sample in each experiment. **E** SIRF detection of MRE11 at normal and stalled replication forks in the same U2OS cells described in (**B**). Cells were pulse labeled with EdU for 8 min and treated with or without 4 mM HU for 3 h. Scale bars: 10 μm. Images with the red channel are provided in Supplementary Fig. 10. Three independent experiments were performed. *P*: One-way ANOVA. Red line: mean. Western blot showing expression of RNAi-resistant S96A, S96D, and WT-STN1. *n* = ~200 cells were analyzed per sample in each experiment. **F** Images showing chromosome aberrations in STN1 knockdown HeLa cells with ectopic expression of S96A, S96D, and WT-STN1. Cells were treated with HU (2 mM, 3 h). Aberrations are labeled with red stars. Examples of aberrant chromosomes are amplified and shown in inserts, with red arrows pointing to aberrations. Two independent experiments were performed, and the result from one experiment is shown. *P*: One-way ANOVA. Source data are provided in the Source Data file.

---

sensitive and quantitative visualization of protein localization at replication forks at a single cell level when the target protein is in close proximity with EdU-labeled nascent strands. Briefly, cells were grown exponentially on chamber slides overnight and pulse labeled with EdU for 8 min, followed by 4 mM HU treatment for 3 h. Using the click reaction, the incorporated EdU is biotinylated and can be recognized by the biotin antibody. The proximity ligation assay (PLA) is then performed to detect the target protein's interaction with the biotinylated EdU at the stalled fork[58] (Fig. 1C). In agreement with our previous observation[29], we observed that WT-STN1 localized at stalled forks (Fig. 1D). In contrast, STN1-ΔIDR drastically reduced STN1 localization at forks (Fig. 1D). Together, our results suggest that the IDR is crucial for STN1 localization at forks and protecting stalled forks from abnormal NSD.

## Phosphorylation of the S96 residue in STN1 IDR controls STN1 localization to stalled forks and is essential for antagonizing NSD

Considering the importance of IDR in STN1, we analyzed the sequences of OB domains of STN1 homologs in higher eukaryotes. We observed that the IDR sequence in STN1 was highly conserved, including a number of serines and threonines such as T94, S96, S98, S108, T110 and S111 (Fig. 2A). Analysis of the Cancer Genomics Atlas (TCGA) and COSMIC database[59,60] showed that four cancer-associated somatic missense mutations, namely E95G (breast invasive ductal carcinoma), S96V (malignant melanoma), V97A (cervical squamous cell carcinoma), and S102T (uterine endometrioid carcinoma), are present in this IDR. Three of them (E95G, S96V, V97A) are located at or adjacent to S96. We therefore focused on the S96 residue. Since IDRs often regulate protein functions via post-translational modifications (PTMs), we suspected that S96 might be subject to PTM. We then mutated S96 to alanine (S96A, phosphor inactive) or aspartic acid (S96D, phosphomimetic) and examined their localization at stalled forks using the SIRF assay. Using retroviral transduction, we constructed stable cell lines expressing Myc-tagged WT-STN1, S96A, and S96D. We observed that S96A localization to stalled forks drastically decreased, while the phosphomimetic mutant showed similar ability of fork localization like WT (Fig. 2B).

Next, we performed the DNA fiber assay to determine whether S96 phosphorylation played a role in antagonizing NSD. The endogenous STN1 was depleted with shRNA in both the U2OS cell line and the normal skin fibroblast BJ immortalized with telomerase (BJ/hTERT), and then the RNAi-resistant S96A, S96D, and WT-STN1 were expressed (Fig. 2C, D). While re-expressing WT-STN1 in STN1-depleted cells completely rescued NSD, the S96A mutant failed to protect forks from NSD in U2OS cells (Fig. 2C, left panel). Treating cells with MRE11 inhibitor mirin reversed the fork degradation in S96A, suggesting that S96 phosphorylation is important for protecting the nascent-strand DNA from aberrant MRE11-mediated degradation (Fig. 2C, left panel). In contrast, the phosphormimetic S96D fully rescued NSD caused by

STN1 depletion (Fig. 2C, right panel). Similar results were observed in BJ/hTERT cells, suggesting that the effect of S96A on fork degradation was not cell line specific (Fig. 2D).

Since CST limits excessive MRE11 nuclease access to stalled forks[29], we performed MRE11 SIRF to measure whether S96 phosphorylation regulated MRE11 localization at stalled forks. Consistent with our previous observation[29], MRE11 localization to stalled forks markedly increased in STN1-depleted cells, and WT-STN1 and S96D effectively blocked MRE11 localization to stalled forks (Fig. 2E). In contrast, S96A was unable to inhibit MRE11 localization to stalled forks, suggesting that S96 phosphorylation is crucial for blocking MRE11 access to stalled forks (Fig. 2E).

Next, we determined the effect of S96A on chromosome stability under the perturbed replication condition. We stably expressed RNAi-resistant WT-STN1, S96D, and S96A in HeLa cells by retroviral transduction and concurrently knocked down endogenous STN1, then treated with or without HU for 3 h, followed by metaphase chromosome spreading. Depleting endogenous STN1 enhanced chromosome abnormalities that were rescued by WT-STN1 or S96D expression, while S96A failed to rescue (Fig. 2F). In line with this, S96D was able to rescue the hypersensitivity to HU caused by STN1 depletion and S96A failed to rescue (Supplementary Fig. 2). Collectively, these results suggest that S96 in the IDR regulates key functions of STN1, including maintaining genomic stability and protecting stalled forks under the perturbed replication condition.

## STN1 S96 phosphorylation is stimulated under perturbed DNA replication conditions

The above results suggest that S96 is likely targeted by post-translational phosphorylation and such phosphorylation is important for STN1 function at forks. We then generated a custom antibody that specifically recognized phosphorylated STN1 at S96 (pS96). We performed western blot analysis on whole cell lysates from HeLa cells expressing Myc-WT-STN1 (WT) or Myc-S96A using the anti-pS96 antibody. It recognized WT-STN1 but not the S96A mutant (Fig. 3A), indicating that the anti-pS96 antibody was specific to pSTN1. Using this antibody, we detected a basal level of S96 phosphorylation under the unperturbed condition (Fig. 3B). Notably, pSTN1 levels increased upon HU or APH treatment in two different cell lines U2OS and HeLa, suggesting that STN1 phosphorylation is stimulated upon fork stalling (Fig. 3B). The anti-pS96 antibody exhibited similar recognition of the phosphomimetic mutant S96D when compared to the WT (Fig. 3C), and phosphatase treatment abolished pS96 signal (Fig. 3C), further validating the specificity of the anti-pS96 antibody.

## STN1 S96 can be phosphorylated by ATR/CHK1 in response to perturbed DNA replication

We then sought to identify the potential protein kinase(s) that phosphorylated S96 in response to replication stress. Search using the kinase

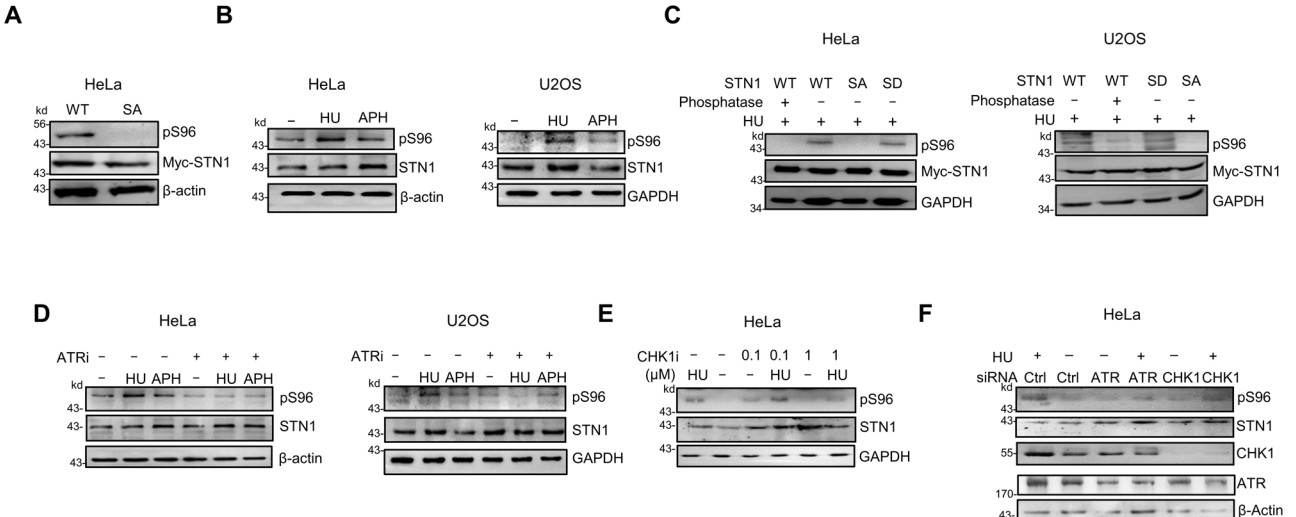

**Fig. 3 | S96 phosphorylation is stimulated by replication stress and ATR-CHK1 phosphorylates S96. A** Western blot was performed to show the specificity of the anti-pS96 antibody. Whole cell lysate of HeLa cells transiently expressing Myc-WT-STN1 or Myc-S96A were used on SDS-PAGE. Anti-pS96 antibody was used to detect S96 phosphorylation of Myc-STN1. Representative blots from three independent experiments are shown. Full blots are provided in Supplementary Fig. S6. **B** Western blot of pS96 of endogenous STN1 protein in HeLa and U2OS cells after the HU (10 mM, 3 h) or APH (5 µg/ml, 3 h). 10 mM HU was used in our initial experiments. In later experiments we used 4 mM HU and observed similar pS96 stimulation (see **C**; Figs. 4 and 7 below). Representative blots from three independent experiments are shown. **C** Western blot of pS96 in HeLa and U2OS cells after the HU treatment (4 mM, 3 h). Whole cell lysates were treated with 100 units of lambda phosphatase

for 30 min at 30 °C prior to loading. Representative blots from three independent experiments are shown. **D** Western blot of pS96 in HeLa and U2OS cells after pretreatment with ATRi (VE821, 20 µM, 3 h) followed by 10 mM HU for 3 h or 5 µg/ml APH for 3 h treatment. Representative blots from three independent experiments are shown. **E** Western blot showing effects of CHK1i on S96 phosphorylation under the replication stress in HeLa cells. HeLa cells were pretreated with 0.1 µM or 1 µM Chk1i (Prexasertib) for 1 h followed by 10 mM HU for 3 h or 5 µg/ml aphidicolin for 3 h. Representative blots from three independent experiments are shown. **F** Western blot of pS96 after ATR and CHK1 knockdown. Fourty eight hours after ATR or CHK1 was depleted by siRNA, cells were treated with 10 mM HU for 3 h, and western blot was performed. Representative blots from three independent experiments are shown. Source data are provided in the Source Data file.

predicting databases like NetPhos-3[61] shows that S96 might be a substrate of several kinases including CaM-II, PKC, and GSK3, but not ATR or CHK1. However, their confidence scores were moderate to low. CaM-II had a confidence score of 0.484, PKC had 0.451, and GSK3 had 0.450. Despite this, considering that the ATR-CHK1-mediated phosphorylation is the major pathway regulating replication stress response[62], we tested whether ATR-CHK1 could phosphorylate S96. HeLa or U2OS cells were pretreated with ATR inhibitor (VE821) prior to HU or APH treatment, and pS96 was measured by western blot. VE821 treatment reduced HU- and APH-induced pS96 phosphorylation in both HeLa and U2OS (Fig. 3D). To determine whether ATR phosphorylated STN1 through CHK1, we pretreated HeLa cells with the CHK1 inhibitor (prexasertib) in two different concentrations, 0.1 µM and 1 µM prior to HU treatment. While we did not observe significant change in S96 phosphorylation at the lower concentration of CHK1i (0.1 µM), the 1 µM treatment effectively reduced pS96 (Fig. 3E). To further validate that STN1 was phosphorylated by ATR and CHK1, we depleted ATR or CHK1 using siRNA and found a substantial decrease of HU-stimulated pS96 phosphorylation upon ATR or CHK1 depletion (Fig. 3F).

## STN1 is phosphorylated by CaMKK2 under replication stress

A second replication stress response pathway is mediated by the CaMKK2-AMPKα pathway[8]. Fork stalling induced by HU causes an elevation of intracellular concentration of $Ca^{2+}$, activating the CaMKK2 kinase that further phosphorylates its downstream target AMPKα. The activated AMPKα then phosphorylates EXO1 to inhibit its nuclease function, therefore protecting stalled forks from aberrant resection by EXO1[8]. We speculated that the activated CaMKK2-AMPKα pathway caused by elevated intracellular $Ca^{2+}$ concentration ($[Ca^{2+}]_i$) could also phosphorylate STN1. We first confirmed that HU or APH treatment caused $Ca^{2+}$ influx under our experimental condition using a GFP-based reporter GCaMP6s, which emits green fluorescence signals once interacting with $Ca^{2+}$. The GFP intensity inside the cell correlates with

the $[Ca^{2+}]_i$ level[63]. Consistent with our previous report[8], HU or APH treatment resulted in enhanced accumulation of GFP signals, indicative of elevated $[Ca^{2+}]_i$ (Fig. 4A).

We then tested whether S96 phosphorylation could be stimulated in response to increased $[Ca^{2+}]_i$. Indeed, elevating $[Ca^{2+}]_i$ using either the calcium ionophore A23187 (causing the influx of extracellular $Ca^{2+}$)[64] or the SERCA inhibitor thapsigargin (causing calcium release from intracellular stores[65,66] stimulated S96 phosphorylation in both HeLa and U2OS cells (Fig. 4B), suggesting that STN1 phosphorylation responds to increased $[Ca^{2+}]_i$. The increase in S96 phosphorylation caused by calcium ionophore A23187 and thapsigargin was abolished by phosphatase treatment, again validating the pS96 antibody specificity (Fig. 4B). To further validate the role of calcium signaling in S96 phosphorylation, we pretreated cells with the cell permeable $Ca^{2+}$ chelator BAPTA-AM for 30 min. Once inside the cell, BAPTA-AM is cleaved by intracellular esterases to release BAPTA, which then binds to and sequesters $Ca^{2+}$ within the cell, effectively reducing the intracellular calcium concentration[67]. BAPTA-AM pretreatment caused a significant reduction in S96 phosphorylation after calcium ionophore A23187 and thapsigargin treatment (Supplementary Fig. 3).

Using two CaMKK2 KO clones we previously generated[8], we found that S96 phosphorylation level was reduced to the basal level in CaMKK2 KO cells treated with HU or A23187, suggesting that CaMKK2 is important for STN1 phosphorylation (Fig. 4C, D). Inhibition of CaMKK2 kinase activity with its inhibitor (STO-609) drastically reduced the STN1 phosphorylation, suggesting that STN1 phosphorylation was mediated by the CaMKK2 kinase activity (Fig. 4E). Since CaMKK2 activates its downstream kinase AMPKα, we then tested whether APMKα was required for S96 phosphorylation. Surprisingly, using two AMPKα KO clones where both AMPKα1 and AMPKα2 genes were deleted by CRISPR/Cas9[8], we observed a moderate increase of S96 phosphorylation, indicating that CaMKK2 phosphorylates STN1 in an AMPKα-independent manner (Fig. 4F).

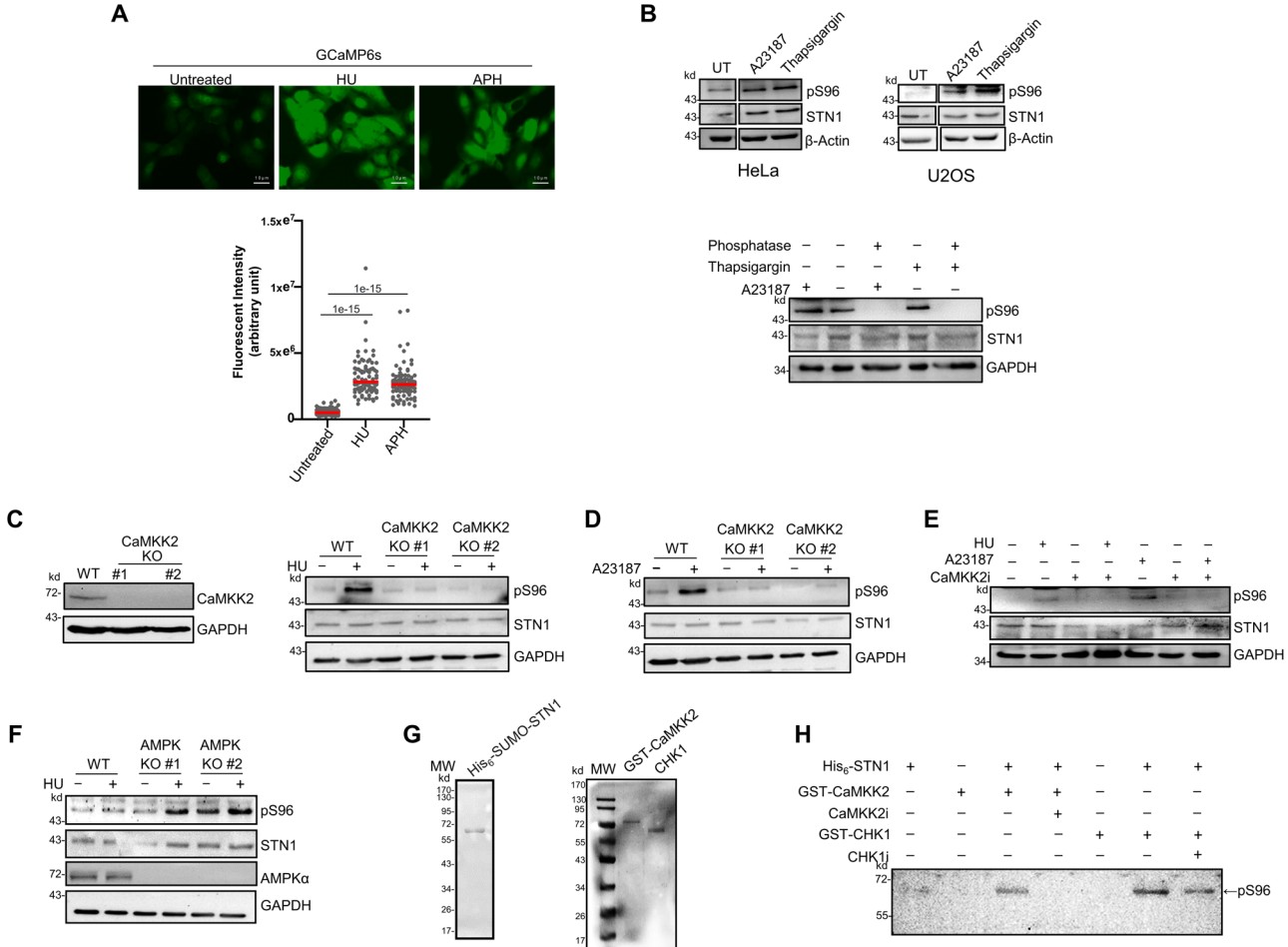

**Fig. 4 | CaMKK2 phosphorylates STN1 S96 in response to Ca²⁺ concentration increase and HU treatment. A** GCaMP6s reporter assay showing the increase of intracellular Ca²⁺ concentration in U2OS cells after treatment with HU (4 mM) or APH (5 μg/ml) for 3 h. Scale bar: 10 μm. Quantification of the images was performed using Image J. *P:* Two-tailed unpaired *t*-test. Red line: mean. Two biologically independent experiments were performed and *n* = -100 cells were analyzed per sample in each experiment. **B** Western blot of pS96 after Ca²⁺ ionophore (A23187, 2 μM, 1 h) or thapsigargin (1 μM, 1 h) treatment in both HeLa and U2OS cells. Images shown in upper panels were cropped from the same blot. For the phosphatase treatment, cell lysates were treated with 100 units of lambda phosphatase for 30 min at 30 °C. Representative blots from three independent experiments are shown. **C** Western blot of pS96 in CaMKK2 KO cells after HU treatment. Two CaMKK2 KO HeLa clones were treated with HU (4 mM, 3 h) and pS96 was detected. Representative blots from two independent experiments are shown. **D** Western blot of pS96 in CaMKK2 KO cells after A23187 treatment. Two CaMKK2 KO HeLa clones were treated with A23187 (2 μM, 1 h). Representative blots from two

independent experiments are shown. **E** CaMKK2 inhibition diminishes pS96. HeLa cells were pretreated with 25 μM CaMKK2i (STO-609) for 30 min followed by 10 mM HU treatment for 3 h or 2 μM A23187 treatment for 1 h. Western blot was performed using whole cell lysate. Representative blots from two independent experiments are shown. **F** Effect of AMPKα KO on S96 phosphorylation. Two AMPKα KO clones were treated with 4 mM HU for 3 h and pS96 was detected. Representative blots from two independent experiments are shown. **G** SDS-PAGE showing purified His₆-SUMO-STN1, GST-CaMKK2 and CHK1 proteins used in the in vitro kinase assay. Representative blots from three independent experiments are shown. **H** In vitro kinase assay. His₆-SUMO-STN1 was incubated with CaMKK2 or CHK1 in the kinase reaction buffer with or without CaMKK2i or CHK1i. STN1 phosphorylation was detected with the anti-pS96 antibody after SDS-PAGE. Three independent experiments were performed and results from one experiment was shown. Representative blots from three independent experiments are shown. Source data are provided in the Source Data file.

## STN1 can be directly phosphorylated by CaMKK2 and CHK1 in vitro

Next, to determine whether CaMKK2 or CHK1 can directly phosphorylate S96, we performed in vitro kinase assay using purified recombinant CHK1 and CaMKK2 kinases and purified recombinant STN1 protein. We first attempted to express human STN1 in *Escherichia coli* to purify the unphosphorylated form of STN1, but found that expressed protein was largely insoluble. We then tagged the full-length STN1 with the SUMO tag to improve its solubility in *E. coli*. The His₆-tagged SUMO-STN1 was purified with immobilized Ni⁺ affinity chromatography (Fig. 4G). Purified CaMKK2 or CHK1 proteins were then directly incubated with the SUMO-STN1 protein, and a western blot was performed to detect S96 phosphorylation using the anti-pS96 antibody. We observed pS96 after the incubation (Fig. 4H), and the addition of

CaMKK2i or CHK1i in the reaction inhibited the S96 phosphorylation, suggesting that phosphorylation was specific to CaMKK2 or CHK1 (Fig. 4H). Together, our results suggest that STN1 can be directly phosphorylated by CHK1 or CaMKK2 at S96 in response to replication stress or increased [Ca²⁺]ᵢ.

## ATR-CHK1 and CaMKK2 independently phosphorylate STN1

As both ATR-CHK1 and CaMKK2 phosphorylate pS96 during perturbed replication (Figs. 3 and 4), we then determined whether these two pathways compete or collaborate in phosphorylating pS96. We treated HeLa and U2OS cells with CHK1 and CaMKK2 inhibitors, both individually and in combination. Co-inhibition of CHK1 and CaMKK2 resulted in a further decrease in S96 phosphorylation under replication stress in both HeLa and U2OS cells, indicating independent

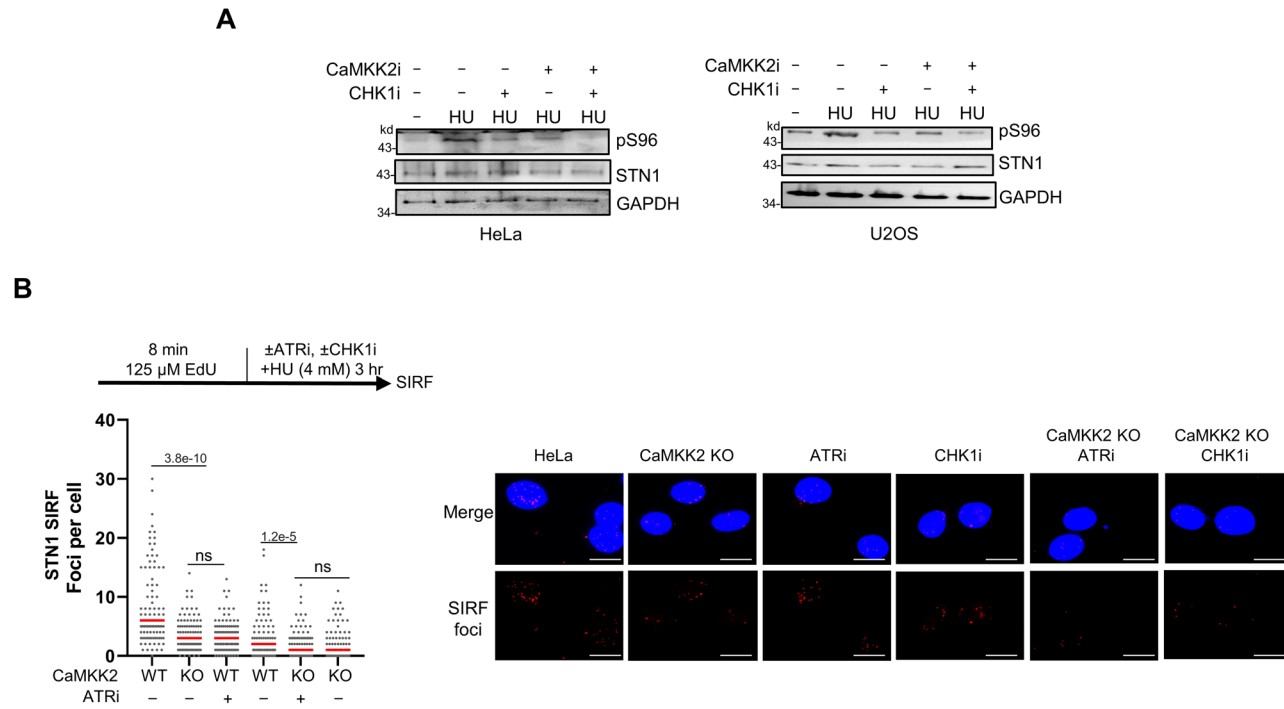

**Fig. 5 | ATR-CHK1 and CaMKK2 phosphorylate STN1 independently. A** Western blot of pS96 in HeLa and U2OS cells after co-inhibition of CHK1i (prexasertib, 1 μM, 1 h), and CaMKK2i (STO-609, 25 μM, 30 min), followed by 4 mM HU for 3 h. Representative blots from two independent experiments are shown. **B** SIRF detection of STN1 after co-inhibition of CHK1i (prexasertib, 1 μM, 1 h), CaMKK2i (STO-609, 25 μM, 30 min), and ATRi (VE821, 20 μM, 3 h). Cells were pulse labeled with EdU for 8 min, followed by 4 mM HU for 3 h. Scale bars: 10 μm. Representative images from two independent experiments are shown. *P*: One-way ANOVA. Red line: mean. *n* = ~100 cells were analyzed per sample in each experiment. Source data are provided in the Source Data file.

contributions from both pathways (Fig. 5A). To determine if the reduced STN1 phosphorylation was a result of a decreased population of replicating cells, we conducted a cell cycle analysis on the treated cells. No significant alteration of S-phase cells was observed after ATR, CHK1, CaMKK2 inhibition, or ATR/CHK1 and CaMKK2 co-inhibition (Supplementary Fig. 4), suggesting that the reduction of STN1 phosphorylation by ATR/CHK1 and CaMKK2 inhibition is unlikely due to the alteration in S phase population.

We then performed STN1 SIRF analysis and observed diminished STN1 localization following ATR inhibition, CHK1 inhibition, or CaMKK2 KO (Fig. 5B). Additionally, simultaneous inhibition of both ATR/CHK1 and CaMKK2 further decreased STN1 localization at stalled forks (Fig. 5B), supporting that CaMKK2 and ATR/CHK1 independently phosphorylate S96.

### S96 phosphorylation is important for RAD51 localization to stalled forks but is dispensable for the CST complex formation, CST interaction with RAD51 and POLα, CST nuclear localization, and ssDNA binding

RAD51 is required for fork reversal and protecting stalled forks from aberrant nucleolytic degradation[9,63,68-70]. We have previously reported that CST interacts with RAD51, and CST depletion significantly reduces HU-induced RAD51 foci formation and RAD51 recruitment to stalled replication forks, which suggest that CST assists RAD51 recruitment to stalled forks[29,46,47]. To determine whether S96 phosphorylation played a role in regulating RAD51 recruitment, SIRF assay was performed in cells co-expressing the RNAi-resistant WT-STN1, S96D, or S96A and the siRNA that depleted endogenous STN1 (Fig. 6A). Consistent with our previous observation[29], we found that STN1 knockdown markedly reduced RAD51 SIRF signal (Fig. 6A). WT-STN1 or S96D expression completely rescued the RAD51 localization to forks, while S96A failed to rescue RAD51 recruitment, indicating that S96 phosphorylation

plays an important role in promoting RAD51 recruitment to stalled forks (Fig. 6A). In agreement with the SIRF results, we found that WT-STN1 and S96D fully rescued the HU-induced RAD51 foci formation, while S96A failed to rescue the RAD51 foci formation (Fig. 6B). As shown by the EdU incorporation assay, S96D, S96A and WT-STN1 showed similar EdU incorporation, indicating that the S-phase population were minimally affected by S96 phosphorylation and the impaired RAD51 recruitment to forks was not due to changes in S phase cells (Supplementary Fig. 5).

It has been reported that along with CTC1 and TEN1, the N-terminus of STN1 interacts with POLα–primase to perform the de novo DNA synthesis[71,72], and CST-POLα interaction regulates nuclear localization of CST[73]. To determine whether S96 phosphorylation affects CST-POLα interaction and STN1 nuclear localization, we first performed co-IP. HEK293T cells were transiently transfected with Myc-tagged WT-STN1, S96D, or S96A (Fig. 6C). No alteration was observed in STN1-POLα interaction with WT-STN1, S96D, and S96A, indicating that S96 phosphorylation does not play a role in regulating STN1-POLα interaction (Fig. 6C). Since the anti-pS96 antibody showed a strong non-specific recognition of an unknown protein (Supplementary Fig. 6) which prevented us from using this antibody to detect pS96 localization with immunostaining, we isolated nuclear and cytoplasmic fractionations to check whether S96 phosphorylation affected STN1 nuclear localization. As shown in Fig. 6D, S96A retained its nuclear localization similar to WT-STN1 (Fig. 6D). Together, these results indicate that STN1 phosphorylation plays a minor role in regulating STN1-POLα interaction and its nuclear localization.

We have previously shown that RAD51 physically interacts with the CST complex after HU or APH treatment and proposed that the CST-RAD51 interaction facilitates RAD51 recruitment to stalled forks[46,47]. To understand how S96 phosphorylation regulates RAD51 recruitment to forks, we first used co-IP to determine the effect of

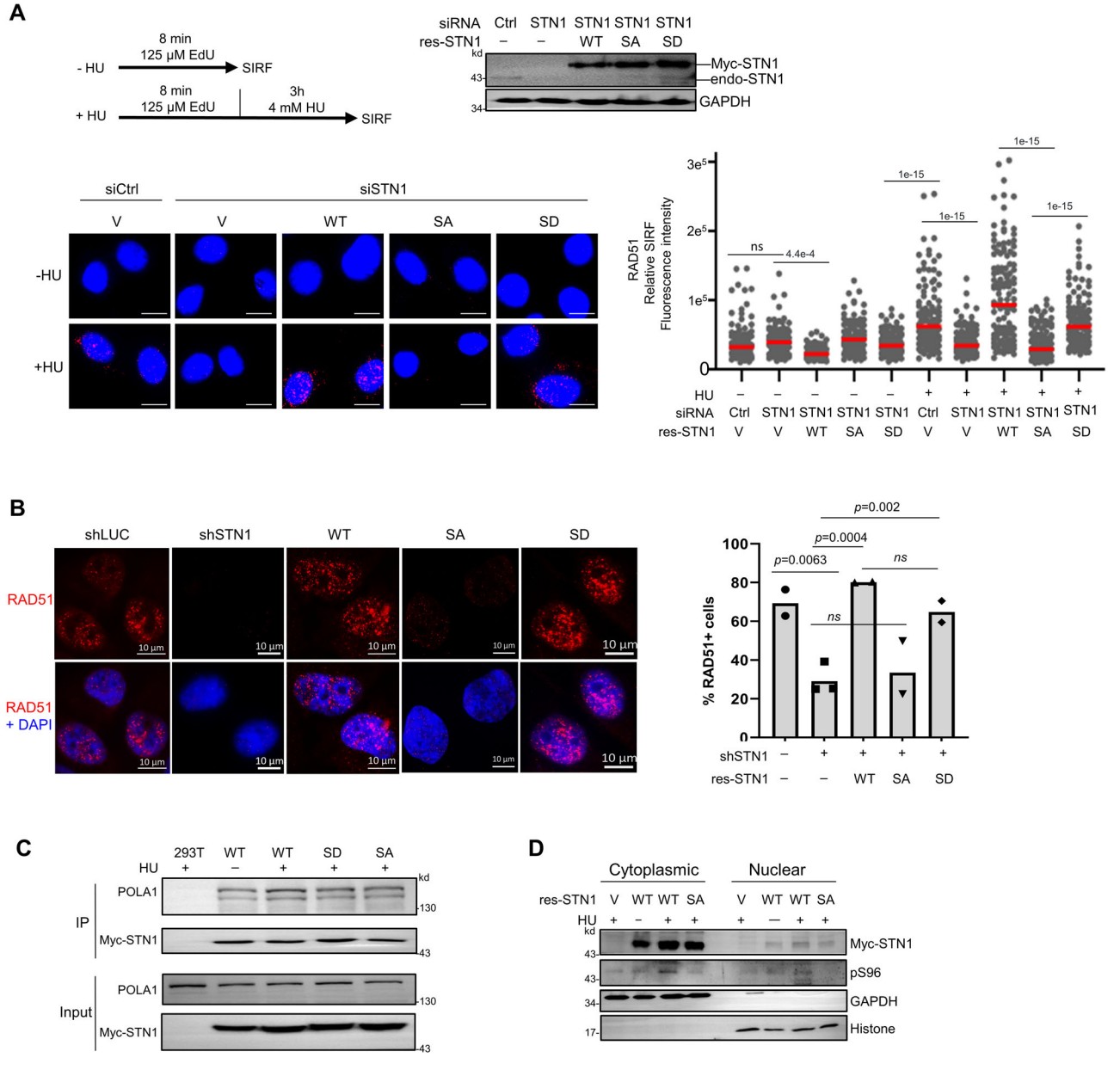

**Fig. 6 | S96A mutation abolishes RAD51 localization at forks in response to replication stress, but does not impact STN1 interaction with POLα or its nuclear localization. A** RAD51 SIRF at normal and stalled replication forks in U2OS cells expressing RNAi-resistant S96A, S96D, and WT-STN1. Endogenous STN1 was concurrently depleted with siRNA. Scale bars: 10 μm. Images with the red channel are provided in (Supplementary Fig. 11). Representative images from two independent experiments are shown. *P:* One-way ANOVA. Red line: mean. Western blot shows the depletion of endogenous STN1 and the expression of S96A, S96D, and WT-STN1. *n* = ~150 cells were analyzed per sample in each experiment. **B** RAD51 IF in HeLa cells stably expressing RNAi-resistant WT, S96A, and S96D. Endogenous STN1

was concurrently depleted with siRNA. Cells were treated with 2 mM HU for 3 h and fixed with paraformaldehyde for IF. Scale bars: 10 μm. The means from two independent experiments are plotted. Error bars: SEM. *P:* one-way ANOVA with post hoc Tukey. **C** STN1-POLα co-IP. HEK293T transfected with Myc-WT-STN1, Myc-S96D, and Myc-S96A were treated with 4 mM HU for 3 h. Myc beads was used for IP. Two independent experiments were performed. **D** Cytoplasmic and nuclear fractionation of WT-STN1 and S96A from U2OS stably expressing Myc-STN1 or Myc-S96A. Cells were treated with or without 4 mM HU treatment for 3 h. Two independent experiments were performed. Source data are provided in the Source Data file.

S96A on the CST complex formation and RAD51 interaction with CST. Interestingly, we found that S96A retained the ability to form the CST complex and interact with RAD51 (Fig. 7A).

Previously we have purified the CST complex from HEK293 cells and reconstituted an in vitro MRE11 degradation assay using purified human MRE11 protein and a DNA substrate mimicking the reversed fork, and shown that the CST complex binding to DNA can directly block MRE11 degradation of DNA[29]. We then tested whether S96 phosphorylation affected the CST binding to DNA and MRE11

degradation using these in vitro assays. We co-expressed CTC1, S96A, TEN1 or CTC1, S96D, TEN1 in HEK293 cells and purified the CTC1-S96A-TEN1 and CTC1-S96D-TEN1 complexes using the methods described previously[29,47] (Fig. 7B). EMSA assay showed that both the CTC1-S96A-TEN1 and the CTC1-S96D-TEN1 complexes exhibited DNA binding affinity comparable to the WT-CST (Fig. 7C), suggesting that S96 phosphorylation plays no role in regulating the intrinsic DNA-binding activity of CST. In agreement with our co-IP data, both the phosphor-inactive and the phosphormimetic CST interacted with RAD51 like WT

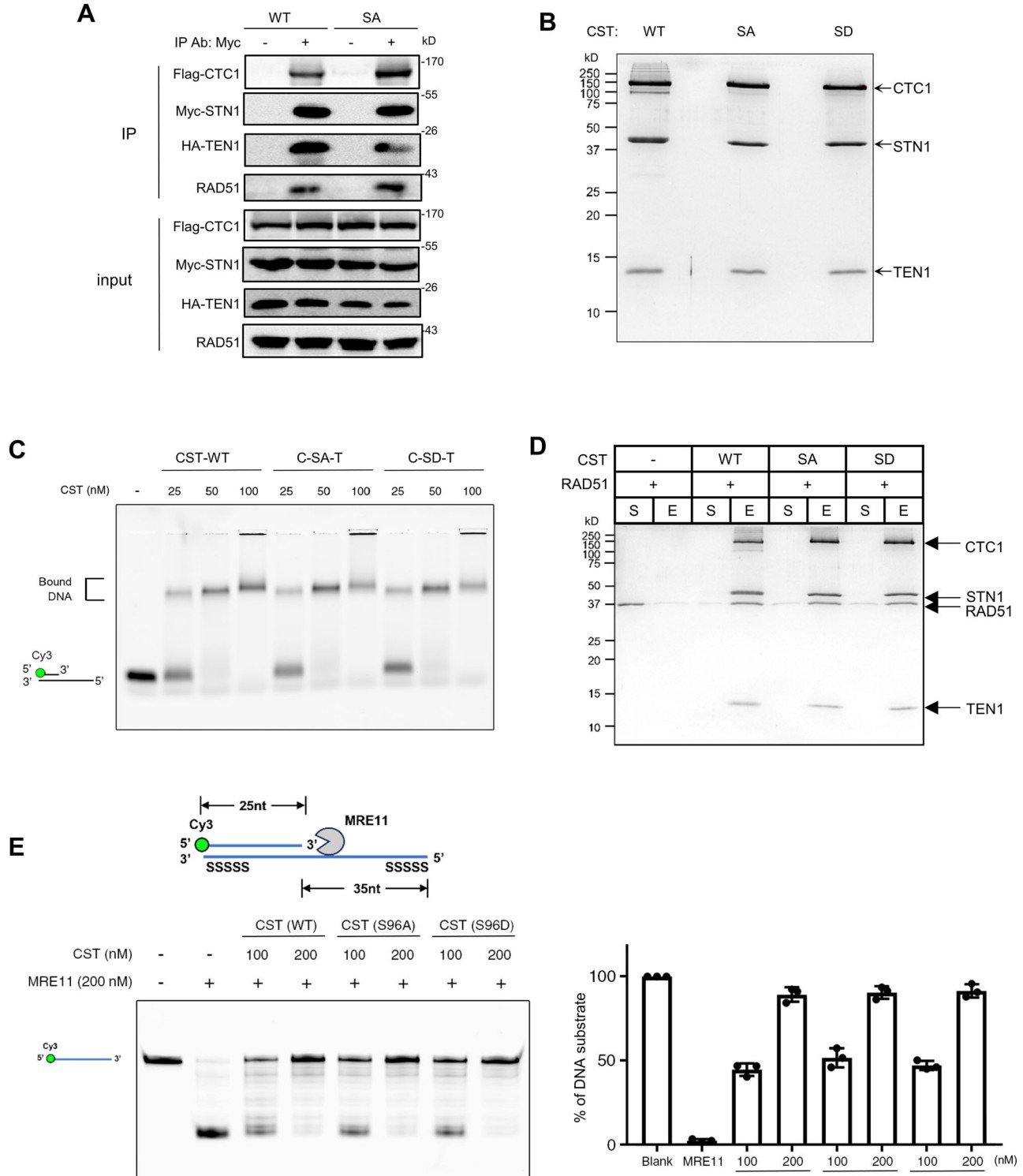

(Fig. 7D). In addition, both the CTC1-S96A-TEN1 and CTC1-S96D-TEN1 complexes were capable of inhibiting MRE11 degradation of DNA in vitro (Fig. 7E). Collectively, our results suggest that while S96 phosphorylation promotes efficient RAD51 recruitment to stalled forks and is critical for antagonizing NSD in cells (Fig. 2), it does not regulate CST-RAD51 physical interaction or CST binding to DNA, suggesting that the S96 phosphorylation may regulate a to-be-identified molecular function of CST that is involved in recruiting RAD51 to stalled forks.

## Cancer-associated mutations impair STN1 phosphorylation and fork stability

As mentioned above, three cancer-associated somatic missense mutations, E95G, S96V and V97A are located at or adjacent to S96. Previously we found that E95G or S96V expression caused chromosome instabilities that was elevated by replication stress and also diminished HU-induced RAD51 foci formation[55]. We suspected that S96 phosphorylation was affected by these mutations, resulting in NSD. Indeed, we found that both E95G and S96V impaired S96

**Fig. 7 | In vitro, S96 phosphorylation has no impact on CST complex formation, CST interaction with RAD51, binding to DNA, or inhibiting MRE11 degradation. A** Co-IP of S96A with CTC1, TEN1, and RAD51 in HEK293T cells co-transfected with Flag-CTC1, HA-TEN1, and MyC-WT-STN1 or Myc-S96A. Cells were treated with 2 mM HU for 3 h. Myc antibody was used for IP. Three independent experiments were performed to ensure reproducibillity. **B** Purified wild-type CST (WT), C-S96A-T (SA), and C-S96D-T (SD) complexes were resolved in 15% SDS-PAGE and stained with Coomassie blue. Representative result from three independent experiments is shown. **C** The DNA-binding ability of the CST complex (WT, SA, and SD) was determined by EMSA. The 5′ Cy3- labeled substrates were incubated with the indicated concentrations of CST. Samples were analyzed with 0.8 % agarose gel. Representative result from three independent experiments is shown. **D** Effects of S96A and S96D on interaction with RAD51 in vitro. Flag-CTC1-STN1-TEN1-His$_6$ (CST- WT, SA, and SD) was incubated with RAD51, followed by incubation with His-Tag Dynabeads to capture the CST and associated proteins using a magnetic bead separator. The supernatant (S) and eluate (E) were analyzed by 15% SDS-PAGE with Coomassie blue staining. RAD51 alone is shown as a control. Three independent experiments were performed and the result from one experiment is shown. **E** Effects of S96A and S96D on protecting DNA from MRE11 degradation in vitro. The scheme shows the nuclease activity of MRE11 in degrading 5′ Cy3-labeled substrates (25 nt + 60 nt ssDNA with phosphorothioate bonds on both ends). 5′ Cy3-labeled substrates were pre-incubated with indicated concentrations of CST (CST-WT, SA or SD), then the reactions were completed by adding MRE11. Samples were resolved in 27% denatured polyacrylamide gel. Images show the representative results of 3 independent experiments. The graph represents mean ± S.D ($n = 3$). Source data are provided in the Source Data file.

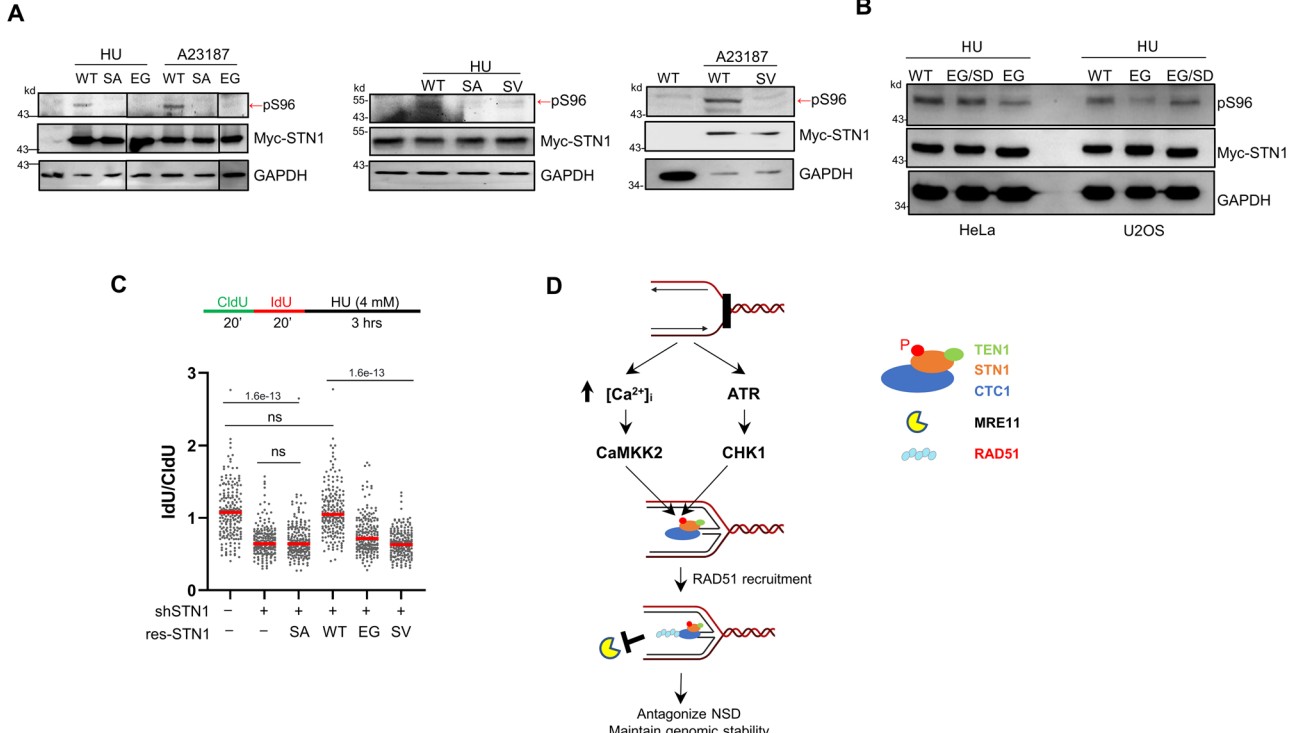

**Fig. 8 | Cancer-associated STN1 mutations impair STN1 phosphorylation and fork stability. A** Western blot of S96 phosphorylation in HeLa cells expressing RNAi-resistant Myc-WT-STN1, Myc-S96V, Myc-S96A, Myc-E95G with concurrent depletion of endogenous STN1. Cells were treated with 4 mM HU for 3 h or 2 μM A23187 for 1 h. Images are cropped from the same blot. Representative blots from three independent experiments are shown. **B** Western blot of S96 phosphorylation in HeLa and U2OS cells expressing Myc-WT-STN1, Myc-E95G, or Myc-E95G/S96D. Cells were treated with 4 mM HU for 3 h. Representative blots from three independent experiments are shown. **C** DNA fiber assay in HeLa cells expressing RNAi- resistant WT-STN1, S96A, S96V, and E95G with concurrent depletion of endogen- ous STN1. Two independent experiments were performed and results from one experiment are shown. *P:* One-way ANOVA. Red line: mean. $n = 200$ fibers were measured per sample in each experiment. **D** Model. Fork stalling leads to the increase of intracellular calcium concentration and accumulation of ssDNA, acti- vating the CaMKK2 and ATR pathways, respectively. Both CaMKK2 and ATR-CHK1 phosphorylate STN1 to promote CST access to stalled forks, which facilitates RAD51 recruitment to stalled forks and blocks MRE11-mediated NSD. Source data are provided in the Source Data file. The model was created with BioRender.com.

phosphorylation (Fig. 8A). An alternative explanation for the dimin- ished recognition of E95G by the anti-pS96 could be that the E95 residue might be a component of the epitope recognized by the pS96 antibody and played a role in antibody binding. Consequently, the E95G mutation could potentially interfere with antibody binding while not impacting S96 phosphorylation. To test this possibility, we engi- neered the E95G/S96D mutant. As shown in Fig. 8B, the pS96 antibody recognized E95G/S96D similarly to WT-STN1, suggesting that the diminished S96 phosphorylation observed in E95G mutant is not due to the interference of antibody binding to the epitope (Fig. 8B). Fur- thermore, cell cycle analysis showed that the S96V and E95G variants had no impact on the cell cycle profile (Supplementary Fig. 7), thus eliminating the possibility that reduced S96 phosphorylation was due to variations in S-phase cell populations.

We then transiently expressed E95G, S96V, WT-STN1, and S96A in cells depleted of endogenous STN1, and treated cells with or without HU. DNA fiber assays showed that neither S96V nor E95G was able to rescue fork degradation caused by STN1 depletion (Fig. 8C). Together, these results suggest the potential role of S96 phosphorylation in cancer development and/or progression.

## Discussion

Protection of stalled forks from aberrant nucleolytic attacks is crucial for maintaining genomic stability. A major signaling pathway

responding to fork stalling in order to stabilize and protect stalled forks is the well-described ATR-CHK1 pathway, which responds to RPA binding to ssDNA accumulated at stalled forks. Recently, we have discovered that stalled replication also activates the cytosolic DNA-induced, $Ca^{2+}$-sensing CaMKK2-AMPKα cascade to protect fork stability via phosphorylating the EXO1 exonuclease, and such phosphorylation inhibits the nuclease activity of EXO1 to prevent the uncontrolled fork degradation by EXO1[7,8]. To date, EXO1 is the only known target of the CaMKK2-AMPKα pathway. It is unknown whether the CaMKK2 signaling pathway also controls other major nucleases like MRE11 and DNA2 that attack reversed forks and degrade nascent strand DNA. In this study, we provide evidence showing that both ATR-CHK1 and CaMKK2 can phosphorylate the human STN1 protein, a component of the CST complex that is recruited to stalled forks and prevents unscheduled MRE11 degradation of nascent DNA at forks[29], at the S96 residue within the IDR of STN1. We show that the S96 phosphorylation is elevated upon HU treatment or calcium influx (Figs. 3 and 4), and such phosphorylation is required for STN1 localizing at stalled forks, inhibiting NSD, and blocking MRE11 access to stalled forks (Fig. 2). Loss of S96 phosphorylation causes an increase in chromosome instability (Fig. 2). In vitro kinase assay shows that STN1 is likely a direct substrate of CaMKK2 and CHK1, as incubating purified CaMKK2 or CHK1 with the STN1 protein leads to S96 phosphorylation (Fig. 4H). Thus, both the ATR and the CaMKK2 signaling pathways phosphorylate STN1 to protect stalled forks from unscheduled NSD (model in Fig. 8D). Interestingly, AMPKα knockout does not abolish STN1 phosphorylation, suggesting that STN1 phosphorylation is distinct from the CaMKK2-AMPKα-EXO1 cascade and is AMPKα-independent (Fig. 4E). Our study indicates that CaMKK2 is able to bypass AMPKα and other downstream kinases (for example, CAMK1 and CAMK4) to directly phosphorylate its targets in response to replication stress. Together, these data identify STN1 as a phosphorylation target of ATR-CHK1 and CaMKK2 signaling pathways in response to replication stress, and suggest that STN1 phosphorylation is required for CST localization at stalled forks in order to recruit RAD51 to stalled forks while limiting MRE11 access to forks (model in Fig. 8D). Furthermore, our findings suggest that the CaMKK2 calcium signaling pathway may play a dual role in both activating fork protectors and inhibiting fork degrading nuclease activities. These results further underscore the critical role of this pathway in maintaining fork stability.

Our results show that CHK1 and CaMKK2 phosphorylate STN1 at the same residue (S96), and these two pathways appear to work independently (Fig. 5). Similarly, CHK1 and AMPKα target the same residue on EXO1 (S746)[8]. It seems that ATR-CHK1 and CaMKK2 are two parallel and co-existing pathways responding to ssDNA accumulation and $Ca^{2+}$ concentration, respectively, since HU-induced CHK1 phosphorylation is unaffected by CaMKK2 KO[8]. At present, it is unclear whether the two pathways compete with each other or collaboratively phosphorylate their targets. It is also unknown how the two independent kinases access the same residue on the same protein. There is a possibility that phosphorylation of S96 on the STN1 protein might promote immediate fork protection. This is because AMPKα is not required for STN1 phosphorylation, and CaMKK2 can directly phosphorylate S96 (Fig. 4H). This action helps prevent the formation of aberrant ssDNA and ensures fork stability during replication stress. On the other hand, EXO1 phosphorylation might regulate long-term responses to such stress by preventing aberrant resection at reversed forks, which ensures the integrity of forks and enables fork restart[8]. Together, these two phosphorylation events might work in concert to provide comprehensive protection to replication forks under replication stress. In addition, it is possible that some replication stress response proteins may be exclusively phosphorylated by one pathway but not by the other. Conversely, some proteins may need both ATR-CHK1 and CaMKK2 to become fully activated. Apart from

STN1 and EXO1, it is highly likely that the $Ca^{2+}$-sensing CaMKK2 signaling pathway also regulates other key factors to protect genome stability under replication stress. Further investigation is needed to identify these downstream factors and to understand the regulatory role of the $Ca^{2+}$-sensing pathway in protecting fork stability and rescuing stalled replication. These studies will not only provide answers to these questions but also help us attain an accurate understanding of how cells respond to replication stress to protect genome stability.

The human CST complex shows structural similarities with the RPA complex[30,74]. Despite such similarity, the two complexes have distinct functions at stalled forks and DSBs. CST protects the nascent strand degradation, while RPA's ability to antagonize nascent strand DNA degradation is weak[69]. At DSBs, CST inhibits DSB end resection[75], while RPA promotes it[76]. In this study, we reveal that the 26 aa IDR unique to STN1 is essential for stalled fork protection. Notably, previous studies have implicated IDRs in fork protection. One study reports that a conserved cluster of ATM/ATR consensus SQ motifs in the IDR of mouse RIF1 is phosphorylated in proliferating B lymphocytes. Such phosphorylation is crucial to protecting NSD from DNA2 nuclease under replication stress[77]. Another report shows that CHK1 phosphorylation of the S255 residue in the IDR of PRIMPOL is essential for its priming activity[78], suggesting the importance of IDR in replication stress tolerance pathways to promote optimal cellular outcomes. These studies and ours suggest that phosphorylation of key amino acids in IDRs is important in regulating replication rescue pathways. It is worth noting that other important fork protectors such as BRCA2 and BRCA1 also contain IDRs[79,80]. It would be interesting to investigate whether these IDRs are involved in fork protection and maintaining genomic stability.

Our cell-based assays reveal that S96 phosphorylation is critical for RAD51 localization to stalled forks and preventing excessive MRE11 access to forks, suggesting that S96 phosphorylation is essential for recruiting RAD51 to stalled fork (Fig. 6). Previously we have reported that CST interacts with RAD51 and assists RAD51 recruitment to stalled forks. CST depletion reduces RAD51 localization to stalled forks, and CST's ability to recruit RAD51 depends on its ability to bind to DNA and interact with RAD51[29,47]. Surprisingly, our co-IP and EMSA results show that S96 phosphorylation has no obvious impact on the CST complex formation, binding to DNA, or interaction with RAD51 (Figs. 6 and 7). In line with this notion, the in vitro MRE11 degradation assay shows that both CTC1-S96A-TEN1 and CTC1-S96D-TEN1 are fully capable of inhibiting MRE11 degradation similar to WT-CST (Fig. 7E). These results indicate that STN1 phosphorylation, while not regulating CST binding to ssDNA and interacting with RAD51 in vitro, is essential for CST to assist RAD51 localization to stalled forks and antagonize NSD in cells, likely regulated by some other unknown mechanisms. We notice that while S96A is still able to migrate into the nucleus and interacts with POLα, it shows reduced localization at stalled forks (Figs. 2B and 5B), indicating that S96 phosphorylation specifically regulates the fork localization of STN1 but not its nuclear localization. We speculate that an unknown protein may interact with and facilitate CST localization to stalled forks and assist CST to recruit RAD51. Such interaction may require S96 phosphorylation. It should be noted that the S96 residue resides in the IDR of STN1 (Fig. 2A). Owing to their conformational flexibility, IDRs are commonly involved in protein–protein interactions[81,82]. As IDRs are rich in polar amino acids, many residues in IDRs are prone to PTMs which may regulate key functions of a protein, for example, interaction with other signaling molecules, protein folding, etc. and consequently, changes protein functions in various biological settings[33,83]. The STN1 IDR is enriched with serines, threonines, and lysines that are easily accessible to PTMs (Fig. 2A). It is possible that apart from S96, other polar residues may also be phosphorylated and are involved in protein-protein interactions. Further investigation is needed to test this possibility.

While this study focuses on understanding the role of S96 phosphorylation in antagonizing NSD and fork protection, it remains a possibility that S96 phosphorylation may play a role in regulating CST functions in telomere maintenance, DSB repair, and replication origin firing and licensing. In fact, a basal level of STN1 phosphorylation can be detected by the anti-pS96 antibody (Fig. 3), indicating that such phosphorylation may be important for normal DNA metabolism under unchallenged conditions, perhaps in facilitating the efficient replication of difficult-to-replicate sequences such as telomeres.

The TCGA database shows that CST genes are altered in various types of cancers[59,60,85,86]. Using a conditional knockout mouse model, we have recently shown that STN1 deficiency promotes colorectal cancer development in young adult mice[45], implicating STN1 and perhaps the CST complex in carcinogenesis. The loss of S96 phosphorylation and increased NSD caused by the cancer-associated somatic mutations (Fig. 8) suggest that STN1 phosphorylation may play a role in the tumor development process. It will be interesting to investigate whether impaired STN1 phosphorylation promotes tumor formation and whether it is possible to specifically target STN1 phosphorylation to facilitate cancer therapy.

## Methods

### Cell culture
HeLa, U2OS, and BJ/hTERT cells were obtained from American Type Culture Collection (ATCC). CaMKK2 KO and AMPKα KO clones were described previously[8]. All cells were cultured in Dulbecco's Modified Eagle's Medium (DMEM, GE Healthcare Life Sciences, Logan, UT, USA) with 10% cosmic calf serum (Hyclone) or fetal bovin serum (Atlanta Biologicals) at 37 °C with 5% $CO_2$. All cell lines were authenticated using the short tandem repeat method. Plasmid transfection was performed using polyethyleneimin (PEI).

### Plasmids
RNAi-resistant pBabe-puro-Myc-STN1 was described previously[46]. pBabe-puro-Myc-S96A, pBabe-puro-Myc-S96D, pBabe-puro-Myc-E95G, pBabe-puro-Myc-S96V, pBabe-puro-Myc-E95G/S96D were generated using the QuikChange II site-directed mutagenesis kit (Agilent, Santa Clara, CA, USA) and then sequenced to confirm that no other mutations were introduced. pcDNA-Flag-CTC1, pcDNA-HA-TEN1, and pCI-neo-Myc-STN1 were described previously[46]. The pcDNA3.4-Flag-MRE11-His$_6$ was used for the expression of human MRE11 protein, and the pEAK8 Flag-CTC1, pcDNA3.4-STN1-TEN1-His$_6$ with a dual CMV promoter were used for the co-expression of CST complex as described previously[29]. The STN1 S96A and S96D expression constructs were generated with site-directed mutagenesis by using pcDNA3.4-STN1-TEN1-His$_6$ as the template. All constructs were sequenced to ensure sequence accuracy.

### RNAi
Small interference RNA (siRNA) sequence targeting human STN1 (targeting GCTTAACCTCACAACTTAA) was described in our previous studies[29,46,50]. siRNAs targeting ATR and CHK1 (targeting TTTGGTAAAGAATCGTGTC) were described previously[8]. Control siRNA sequence was AATTCTCCGAACGTGTCACGT. Cells were transfected with siRNA oligos at a final concentration of 20 nM using Xtreme RNAi transfection reagent (Sigma) according to the manufacturer's protocol. Cells were collected 48 h post transfection for western blotting analysis.

Cells expressing stable shRNAs were generated by retroviral transduction followed by puromycin selection. Control shRNA targeting luciferase was CGUACGCGGAAUACUUCGA (shLUC).

### Small molecules
The following small molecules were used: hydroxy urea (Thermo Fisher Scientific, A10831), aphidicolin (Sigma, 38966-21-1), calcium ionophore A23187 (Sigma, C4403), STO-609 (ApexBio, B6787), VE821 (Sigma, SML1415), BAPTA-AM (Cayman chemical, 15551), EdU (Lumiprobe, 10540), BrdU (Sigma, B23151), propidium iodide (MP biomedicals, SKU:02195458-CF) and Prexasertib (Selleckchem, LY2606368).

### Western blotting and antibodies
Western blotting was performed as described[87]. Briefly, cells were lysed in 1% CHAPS buffer and resolved on SDS-PAGE, transferred to the PVDF membrane, blocked in 5% non-fat dry milk in 1× TBST for 1 h, and followed by primary antibody incubation at 4 °C overnight. Key western blot experiments, especially for detecting pS96 status, were performed with at least three biological replicates. For lambda phosphatase treatment, cell lysates were treated with lambda phosphatase (Santa Cruz Biotechnologies, sc-200312A) for 30 min at 30 °C. The following primary antibodies were used: anti-STN1 (WB 1:1000, Sigma, WH0079991M1); anti-STN1 (SIRF 1:100, Abcam, ab89250), anti-β-actin (WB 1:60,000, Sigma, A5441), anti-CaMKK2 (WB 1:1000, Santa Cruz, sc-271674), anti-Flag (WB 1:2000, Sigma, F1804), anti-Myc (WB 1:30,000, Bethyl, A190-105), anti-AMPKα (WB 1:1000, Cell signaling, 2532), anti-HA (WB 1:10,000, Bethyl, A190-108), anti-Chk1 (WB 1:2000, Santa cruz, sc-8408), anti-MRE11 (SIRF 1:200, Abcam, ab214), anti-RPA32 (WB 1:1000, Bethyl, A300-244A), anti-RAD51 (IF 1:10,000. SIRF 1:200, Abcam, ab63801), anti-biotin (SIRF 1:100 Millipore-Sigma, SAB4200680), anti-biotin (SIRF 1: 200, Cell Signaling, 5597), anti-GAPDH (WB 1:1000, Cell Signaling, 5174), anti-POLα (WB 1:2000, Bethyl, A302-851), anti-histone H3 (WB 1:3000, Active motif, 61277), anti-pAMPKα-Thr172, (WB 1:1000, Cell Signaling, 2531), ATR (WB 1:1000, Cell Signaling, 2790), anti-BrdU (IF 1:200, Roche, 11170376001).

Secondary antibodies: HRP goat anti-rabbit (WB 1:10,0000, Vector Laboratory, PI-1000), HRP goat anti-mouse (WB 1:5000, BD Pharmingen, 554002), goat anti-Mouse Alexa Fluor 488 (IF 1:1000, ThermoFisher, A11029), goat anti-rabbit Alexa Fluor 550 (IF 1:1000, ThermoFisher, 84541), Goat anti-rat Alexa Fluor 488 (IF 1:1000, ThermoFisher, AB_2534074).

The anti-pS96 antibody was custom generated by immunizing three rabbits with phosphor-peptide KLNTE(pS)VSAAPS by AbClonal Technology (WB 1:1000, Woburn, WA). Antibody was purified with antigen affinity purification via phosphor peptide and non-phosphor-peptide columns. Antibodies were then tested using cells expressing WT-STN1 and S96A. Antibody with no reactivity with S96A was used throughout the study.

### DNA fiber assay
DNA fiber assay was carried out using the previously published protocol[88]. Briefly, cells were pulse-labeled with the thymidine analogs 50 μM CldU (MP Biomedicals 105478) for 20 min and then 250 μM IdU (Millipore Sigma 54-42-2) for 20 min. Cells were washed with PBS to remove CldU and IdU and further treated with 4 mM HU for 3 h. To inhibit MRE11, mirin (50 μM, Sigma, M9948) was added concomitantly with HU. After harvesting, the cells were lysed in 12 μl lysis buffer (200 mM Tris-HCl pH 7.5, 50 mM EDTA, 0.5% SDS) on glass slides. Slides were then put in a humidified chamber for 2 min for efficient lysis. Slides were then inclined to 15° to allow genomic DNA spread. Spread DNA fibers were air dried for about 30 min and then fixed in methanol: acidic acid (3:1) for 10 min. Slides were then immersed into 2.5 M HCl for 100 min to denature DNA and then washed with 1× PBS three times. Slides were then blocked with 5% BSA for 30 min, immunostained with anti-CldU (Abcam, ab6326) and anti-IdU (BD Biosciences, 347580) antibodies for 1 h in a humid chamber at 37 °C. After washing with PBS three times, slides were incubated with secondary antibodies anti-rat Alexa Fluor 488 (Thermo Fisher Scientific, A11006) and anti-mouse Alexa Fluor 568 (Thermo Fisher Scientific, A11031) at 37 °C for 1 h. After washing with PBS and air dry at room temperature in dark, slides were mounted in the mounting medium without DAPI (Vector Laboratories, H-1000) using coverslips. Images were acquired

using the Zeiss AxioImager M2 epifluorescence microscope at ×40 magnification. DNA tract lengths were analyzed by the ZEN software (Zeiss, Carl Zeiss AG). About 200 fibers were analyzed for each sample in each experiment and data were analyzed by GraphPad Prism. The images for critical experiments were analyzed independently by two individuals in the lab to avoid human bias. To ensure reproducibility, three independent experiments were performed.

### SIRF assay
SIRF assays were carried out as described previously[29,58]. Exponentially growing cells were seeded on the chamber slides and labeled with 125 μM EdU for 8 min, washed with PBS and treated with 4 mM HU for 3 h to induce replication stress. Cells were then pre-permeabilized with 0.25% Triton X-100 for 2 min, followed by 2% paraformaldehyde (PFA) fixation for 15 min at room temperature. Chamber dividers were then removed, cells were washed in PBS three times at room temperature, followed by treatment with 0.25% Triton X-100 for 15 min at room temperature. Click reaction (2 mM copper sulfate, 10 μM biotin azide, and 100 mM sodium ascorbate) was then performed in a humidified chamber for 1 h at 37 °C, followed by washing with PBS three times. In a humidified chamber, cells were blocked with the blocking buffer (10% BSA, 0.1% Triton X-100) at 37 °C for 1 h followed by incubation with primary antibodies at 4 °C overnight. PLA reactions were then performed using Duolink in situ detection reagents red (DUO92008-100RXN, Sigma-Aldrich) following the manufacturer's protocol. The slides were air dried and mounted with mounting media containing DAPI (Vector Laboratories, H-1200). Images were acquired using the Zeiss AxioImager M2 epifluorescence microscope at 40X magnification. Images were analyzed using the ZEN software (Carl Zeiss AG) and graphs were plotted with GraphPad Prism. The images were analyzed independently by 2 individuals in the lab to avoid human bias. To ensure reproducibility, three independent experiments were performed.

### Co-immunoprecipitation (co-IP)
HEK293T cells were co-transfected with Flag-CTC1, HA-TEN1, Myc-WT-STN1, or Myc-S96A-STN1 and treated with 4 mM HU for 3 h. Cells were then lysed in lysis buffer (0.1% NP-40, 50 mM Tris-HCl, pH 7.4, 50 mM NaCl, 2 mM DTT, protease inhibitor cocktail (Roche, 11836170001) followed by sonication three times with 5 s pulses with 1-min intervals on ice. After centrifugation, at 16,089 × g for 10 min at 4 °C, the precleared lysate was immunoprecipitated overnight using the anti-Myc antibody (Santa Cruz, sc-40) at 4 °C with constant rotation. Myc IP pulldown was carried out using protein G conjugated agarose beads. For STN1-POLα co-IP, HEK293T cells transfected with Myc-STN1/S96D/S96A and anti-c-Myc agarose settled resin (ThermoFisher) were used for co-IP. Beads were spun down briefly at 4 °C, followed by washing with cold lysis buffer three times, with each washing for 10 min at 4 °C. The samples were resuspended in SDS-sample buffer and heated at 95 °C for 5 min, followed by western blot analysis. Three independent experiments were performed to ensure reproducibility.

### Immunofluorescence (IF) staining
IF was carried out as published previously[46]. Briefly, HeLa cells were grown on the chamber slides overnight and then treated with 4 mM HU for 3 h. HeLa cells were then fixed in 4% PFA for 15 min, followed by permeabilization with 0.15% Triton X-100 for 15 min. HeLa cells were then washed with 1X PBS three times 5 min each. Cells were then blocked with 10% BSA at 37 °C for 1 h in a humidified chamber. Cells were incubated overnight at 4 °C with primary antibodies, followed by washing with PBS three times 4 min each. The cells were then incubated with secondary antibodies at room temperature for 1 h, followed by washing with PBS three times 5 min each. Cells were mounted using mounting media containing DAPI (Vector Laboratories). Zeiss AxioImager M2 epifluorescence microscope was used to take the images at 100×.

### His$_6$-SUMO-STN1 expression and purification
Human STN1 cloned into the pSMT3 vector[89] was transformed into Rosetta 2™ E. coli strain. One colony of Rosetta 2™ harboring the plasmid was grown in 500 ml LB medium containing kanamycin (30 μg/ml) until OD600 reached ~0.5. Ethanol (2%) and IPTG (0.1 mM) were then added to the medium, and bacteria were incubated at 16–20 °C overnight with slow shaking. Cells were collected by centrifugation, lysed in buffer E (50 mM Tris-HCl pH 7.5, 250 mM NaCl, 10% glycerol) supplemented with lysozyme 0.2 mg/ml and protease inhibitor (Complete™ EDTA-free tablet, Roche) on ice for 30 min, and then 0.1% Triton X-100 was added and incubated for additional 15 min. The mixture was then loaded to pre-cold MicroFluidizer (Microfluidics) to lyse the bacteria. The cell lysate was then centrifuged at 100,000 × g for 1 h at 4 °C, supernatant was transferred to a new tube, incubated with Ni-NTA resin (Qiagen) for 1 h at 4 °C with rotation, then packed into a column. Flow-through was collected by gravity. Bound protein was eluted stepwise with buffer E with 25 mM imidazole, buffer E with 100 mM imidazole, and buffer E with 300 mM imidazole, followed by SDS-PAGE analysis of each fraction.

### In vitro kinase assay
Purified His$_6$-SUMO-STN1 (100 ng) was incubated with 100 ng of recombinant GST-CaMKK2 protein (Abcam, ab268380) or CHK1 (Abcam, ab60762) in the kinase reaction buffer (5 mM MOPS, pH 7.2, 2.5 mM β-glycerophosphate, 5 mM MgCl$_2$, 1 mM EGTA, 0.4 mM EDTA, 0.05 mM dithiothreitol, and 100 mM ATP) with or without CaMKK2i (STO-609, 100 nM) or CHK1i (Prexasertib, 100 nM or 1 μM) for 30 min at 30 °C. The reaction was stopped by adding 2× SDS-sample buffer and boiling. Samples were then loaded on SDS-PAGE, transferred to a PVDF membrane, and phosphorylated STN1 was detected by western blotting using the anti-pS96 antibody.

### Purification of the wild-type CST complex, CST (STN1 S96A or S96D) mutant variants, and MRE11 recombinant proteins
Human CST wild-type complex and MRE11 protein were purified as described previously[29]. The CST (STN1 S96A or S96D) complex was purified using the same protocol as the wild type. In brief, pEAK8-Flag-CTC1 and pcDNA3.4-STN1 (S96A or S96D)-TEN1-His$_6$ plasmids were co-transfected into Expi293F cells according to the instruction manual from ExpiFectamine 293 kit (Thermo Fisher Scientific). Harvested cells were lysed by sonication and centrifugation at 40,000 × g for 1 h. The cell extract was then fractionated through Ni$^{2+}$ NTA-agarose and anti-Flag M2 affinity purification. The affinity-purified CST complex was further purified using size-exclusion chromatography (Superdex 200 increase 10/300 GL column). The peak fractions were pooled and concentrated, then divided into small aliquots and stored at −80 °C.

### DNA substrates
Fluorescence-labeled 5′ overhang DNA substrate was prepared by annealing the synthetic oligonucleotides described below. 60 nt ssDNA (Oligo 1) with the modified phosphorothioate bond (labeled with an asterisk) at both ends to prevent the non-specific nucleases digestion was purchased from IDT: 5′-A*C*G*C*T*GCCGAATTCTAC-CAGTGCCTTGCTAGGACATCTTTGCCCACCTGCAGGTTC*A*C*C*C*−3′. 25 nt ssDNA (Oligo 2) with the Cy3 fluorescence dye at the 5′ end was purchased from Genomics: 5′-Cy3-GGGTGAACCTGCAGGTGGG-CAAAGA-3′. Briefly, equal amounts of oligonucleotides (Oligo 1 + 2) were mixed in the annealing buffer (50 mM Tris pH 7.5, 10 mM MgCl$_2$, 100 mM NaCl, and 1 mM DTT) and heated at 80 °C for 3 min. The mixed reaction was subsequently transferred to 65 °C for 30 min and cooled down slowly to room temperature. The annealed substrate was purified from a 10% native polyacrylamide gel by electro-elution and filter-dialyzed in an Amicon ultra-4 concentrator (Millipore, NMWL 10 kDa) at 4 °C into TE buffer (10 mM Tris-HCl, pH 8.0, and 0.5 mM EDTA). The substrate concentration was quantified by using absorbance at 260 nm

and the molar extinction coefficients of the substrate with Cy3 calculated by Molbiotools online software.

## Electrophoretic mobility shift assay

The Cy3 fluorescence-labeled 5′ overhang DNA substrate (60 nM) was incubated with indicated amounts of CST complex in 10 μL reaction buffer (35 mM Tris-HCl, pH 7.5, 1 mM DTT, 100 ng/μL BSA, and 50 mM KCl) at 37 °C for 5 min. The reaction mixtures were then electrophoresed on 0.8 % agarose gel with 1× TBE buffer (89 mM Tris, 89 mM borate, and 2 mM EDTA, pH 8) at 100 V for 30 min at 4 °C. Gels were analyzed in an Amersham Typhoon 5 Biomolecule imager (Cytiva) with Amersham Typhoon 2.0 software to detect Cy3 fluorescence signal.

## Affinity pulldown assay

To determine physical protein-protein interactions, 1 μM of CST containing a His$_6$ tag at the C-terminus of TEN1 was incubated with 1 μM RAD51 in 10 μL of reaction buffer (35 mM Tris-HCl pH 7.5, 10% glycerol, 0.01% Igepal, 1 mM 2-mercaptoethanol, 10 mM imidazole, and 50 mM KCl) for 20 min at 37 °C. The sample was then mixed with 2 μL of His-Tag Dynabeads (Invitrogen) for another 20 min at 37 °C to capture CST and associated proteins. The beads were captured using a magnetic beads separator, and the supernatants were kept for further analysis. After washing the beads with 100 μL the same buffer without imidazole, bound proteins were eluted in 15 μL SDS-sample buffer. The supernatants and eluates were analyzed by SDS-PAGE and Coomassie blue staining to determine protein contents by iBright FL1500 Imaging System (Invitrogen).

## MRE11 degradation assay

The Cy3 fluorescence-labeled 5′ overhang DNA substrate (60 nM) was incubated with indicated amounts of CST complex in 10 μL reaction buffer (35 mM Tris-HCl, pH 7.5, 1 mM DTT, 100 ng/μL BSA, 2.5 mM MgCl$_2$, 1 mM ATP, 1 mM MnCl$_2$, and 50 mM KCl) at 37 °C for 5 min, followed by incubation of purified MRE11 (200 nM) at 37 °C for 20 min. Reactions were stopped by incubation with 2.5 μL stop buffer (50 mM EDTA, 0.4% SDS, and 3.2 mg/mL proteinase K) at 37 °C for 15 min. Samples were then mixed with the equal volume 2× denature dye (95% formamide, 0.1% Orange G, 10 mM Tris-HCl, pH 7.5, 1 mM EDTA, and 12% Ficoll PM400), heat denatured at 95 °C for 10 min, and analyzed on 27% denature TBE-Urea-PAGE (7 M Urea) with 1× TBE buffer at 300 V for 40 min at 55 °C. Gels were analyzed in an Amersham Typhoon 5 Biomolecule imager (Cytiva) with Amersham Typhoon 2.0 software to detect Cy3 fluorescence signal. Data were analyzed using the PRISM 7 software and shown as mean ± SD.

## Live cells imaging of Ca$^{2+}$ reporters

For live cell imaging, U2OS cells stably expressing GCaMP6s were treated with HU and APH for 3 h. Images were acquired using the Nikon Ti2E inverted fluorescence microscope with a 40× objective. ImageJ was used to quantify the fluorescence signal.

## Non-denaturing BrdU staining

Endogenous STN1 was knocked down in cells using siSTN1. The next day cells were treated with 10 μM BrdU for 48 h. Prior to PFA fixation, cells were treated with 4 mM HU for 3 h. Cells were then fixed and permeabilized with 0.15% Triton X-100 for 15 min, followed by three 5-min washes with PBS. To block non-specific binding, the cells were treated with 10% BSA at 37 °C for 1 h in a humidified chamber. Following this blocking step, the cells were incubated overnight at 4 °C with an anti-BrdU antibody. Standard immunofluorescence (IF) procedures were then carried out as described above.

## EdU staining

EdU staining was conducted utilizing the Click-iT® EdU Cell Proliferation Assays kit (ThermoFisher). Briefly, cells grown in a chamber slide were exposed to 10 mM EdU for 2 h. Subsequently, cells were fixed with 3.7% formaldehyde for 15 min at room temperature and washed twice with 3% BSA in PBS. Fixed cells were treated with the Click-iT® reaction cocktail for 30 min at room temperature on a rocker in dark. Following this incubation, cells were washed once with 3% BSA in PBS and then exposed to a 30-min incubation with 1× Hoechst® 33342 to stain the nuclei. After two PBS washes, the cells were mounted using a mounting solution without DAPI (Vector Laboratories). For image acquisition, a Zeiss AxioImager M2 epifluorescence microscope was used to capture images with a 40× objective.

## Nuclear and cytoplasmic fractionation

U2OS cells stably expressing pBabe-puro vector, WT-STN1, S96D, and S96A were seeded on 10 cm dishes and allowed to grow overnight. After 4 mM HU treatment for 3 h, cells were collected with centrifugation at 300 × g at 4 °C for 5 min. Cell pellets were rinsed twice with pre-chilled 1× PBS, lysed in cold lysis buffer (0.5% NP-40, 80 mM KCl, 5 mM PIPES, pH 8) for a period of 10 min on ice, after which centrifugation was carried out at 500 × g for another 5 min at 4 °C. Supernatant was collected, while the nuclear pellet was subsequently lysed using nuclear lysis buffer (1% SDS, 25 mM Tris-HCl, pH 8, and 5 mM EDTA) on ice for 10 min before being subjected to centrifugation at 17,350 × g for 15 min at 4 °C. The supernatant obtained was then used for western blot analysis.

## Cell cycle analysis

Propidium iodide (PI) staining was carried out to assess the cell cycle profile using the previously published protocols[90]. Briefly, cells were harvested and washed three times with 1× cold PBS, followed by fixation in 70% ethanol for 30 min at 4 °C. Propidium stain solution containing 40 μg/ml of PI and 20 μg/ml RNase A in PBS was then added to the fixed cells. Cells were incubated for 15 min at room temperature in the dark. Stained cells were then analyzed using flow cytometer (BD FACS CANTO$^{TM}$ II), with a minimum of 50,000 cells counted for each sample. FlowJo software was used to analyze the cell cycle distribution.

## Colony formation assay

Seven hundred fifty cells were seeded per well in 6-well plates before being exposed to various concentrations of HU for a duration of 10 h. Media were then replaced with regular growth media to remove HU. Colonies were fixed 10 days after seeding with a mixture of acetic acid:methanol (1:7), then stained with a 0.5% crystal violet solution. Images were taken using iBright$^{TM}$ CL 1500 Imaging system.

## Reporting summary

Further information on research design is available in the Nature Portfolio Reporting Summary linked to this article.

## Data availability

All data supporting the findings of this study are available within the paper and its Supplementary Information. Source data are provided with this paper.

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

## Acknowledgements

We are grateful for Neal Lue at Weill Cornell Medical College for sharing the pSMT-Hs-STN1 construct and protein purification protocol, Maria Fadri for the initial observation of IDR and S96, Sara Knowles, Kelly Wang, Vikas Gaur, Pau Biak Sang for technical help. This study was supported by NIH R01GM098535 and a Siteman Investment Program grant from Washington University (5124) to Z.Y., National Science and Technology Council (NSTC 111-2326-B-002-019) to P.C., and NIH R01CA234266 and R01GM146376 to W.C. The funding bodies had no role in study design, data collection, data analysis and interpretation of data and in writing the manuscript.

## Author contributions

R.K.J., K.H.L., M.C., Y.W., O.S., S.L. performed experiments. Z.Y., P.C., W.C. supervised personnel. W.C. conceived and supervised the overall study. R.K.J., P.C., W.C. wrote the manuscript.

## Competing interests

The authors declare no competing interests.
