## [Peer Review File · Nature Communications]

CaMKK2 and CHK1 phosphorylate human STN1 in response to replication stress to protect stalled forks from aberrant resectionREVIEWER COMMENTS

Reviewer #1 (Remarks to the Author):

In this study, Jaiswal et al. reveal how the phosphorylation on STN1-S96 regulates the function of the CST complex at replication forks upon replication stress. Phosphorylation of STN1 is induced by replication stress and mediates STN1 fork localization to prevent nascent strand DNA degradation by MRE11. STN1 phosphorylation also promotes RAD51 localization to forks for genome stability. Combining cell-based and in vitro biochemical analysis, the authors show that ATR-CHK1 and CaMKK2 are potential kinases for the phosphorylation. Cancer mutations around the phosphorylation site phenocopy the effect of the phosphor-inactive mutation, implicating a role of STN1 in cancer development.

Overall, the authors provide evidence that S96P-dependent STN1 fork localization is crucial for CST function to protect stall replication forks and promote genome stability. However, the mechanism of how phosphorylation controls STN1 localization was not characterized. Recent structural studies have provided functional insights into how CST complex functions to genome maintenance (PMID: 35830881; PMID: 35578024). It is shown that the N-terminus of STN1 not only interacts with CTC1 and TEN1 for CST complex formation but is also intensively involved in the contacts with the polymerase alpha-primase and DNA template to regulate de novo DNA synthesis. If the authors can investigate whether the role of S96 phosphorylation is linked to polymerase alpha-primase regulation, they may have a chance to provide better conceptual advances for the cellular function of the CST complex.

Major comments:

1. The authors show that ATR-CHK1 and CaMKK2 phosphorylate STN1 (Fig. 4). It is essential to know whether inhibition of ATR-CHK1 and CaMKK2 prevents STN1 localization at stalled forks and increases NSD.
2. It has been shown that nuclear localization of CST is regulated during the cell cycle and by polymerase alpha-primase interaction (PMID: 33731801). It is important to know if STN1 phosphorylation affects CST nuclear localization and polymerase alpha-primase interaction.
3. In Fig. 7, the authors investigated the effect of cancer-associated missense mutations on STN1-S96 phosphorylation by western blots using the phosphor-S96 antibody. The data showed that the STN1-E95G mutant was not recognized by the antibody, so it was concluded that the E95G mutation prevents STN1 phosphorylation. However, an alternative explanation is that E95 is part of the epitope of the antibody and contributes to antibody binding. So E95G mutation can disrupt antibody binding without affecting S96 phosphorylation. To overcome the antibody issue, the authors can try to see if the antibody can recognize STN1-E95G/S96D.

Minor comments:

1. Basal level of STN1 phosphorylation was detected in unperturbed cells and induced by HU, suggesting that STN1 phosphorylation is cell cycle regulated. Thus, it is necessary to check whether ATRi, CHKi, or CaMKK2i treatment affects cell cycle progression (Fig. 3 & 4). Because if kinase inhibitor pre-treatment reduces the S phase population, STN1 phosphorylation can be altered. So the authors need to exclude the possibility that the loss of STN1 phosphorylation is not just a secondary effect of kinase inhibition.
2. The paragraph title "ATR/CHK1 phosphorylates STN1 S96 in response to perturbed DNA replication (Line 228)" seems to be an overstatement.

3. To determine the specificity of S96P antibody, phosphatase treatment on HU sample is recommended, better than S96A sample.

Reviewer #2 (Remarks to the Author):

In this study, Jaiswal et al report that STN1 is phosphorylated by CaMKK2 at its intrinsic disordered region S96. The S96 phosphorylation is physiologically relevant and responsible for its localization at stalled forks. This is an interesting and important finding demonstrated by nice DNA fiber and biochemical assays. Furthermore, the current finding also links to STN1 cancer-associated STN1 variants that confer fork protection defects. Overall, this study is well-designed and executed. There are a few minor concerns that need to be addressed in the current manuscript.

1. The author has proposed a working model, however, this is still remains unclear to me how ATR/CHK1 and CaMKK2 function genetically on STN1. Additionally, the physiological end point is important to present in this study, e.g. whether the STN1 mutation(s) are hypersensitive to HU. Do the STN1 mutations affect cell cycle profile.

2. How is CaMKK2 activated under replication stress? is there any marker/evidence to show CaMKK2 is activated under the experimental conditions?

3. Figure 3E: the control for ATR protein level is missing

4. What's the explanation for the KU80 level for A23187 treated cells?

5. Loading controls were missing in multiple blots.

6. Substandard data quality for several of the western blots, which include Figures 1B, 4F, and 7A.

Immunofluorescence images are also over-saturated and some of the scale bars are inconsistent and low resolution.

7. There are typos across the manuscript.

Reviewer #3 (Remarks to the Author):

The authors describe STN1 activation in the CTC1-STN1-TEN1 (CST) complex by the ATR-CHK1 and CaMKK2-AMPK signaling pathways in response to replication stress. They show that replication stress induces STN1 phosphorylation in its intrinsic region by both the ATR-CHK1 and the calcium-sensing kinase CamKK2, and an absence of STN1 activation leads to MRE11-mediated nascent strand degradation and overall genome instability. Although many ATR-CHK1 targets have been identified at stalled forks, so far EXO1 has been the only known target in the CaMKK2-AMPK signaling pathway (Li et al., 2019, Molecular Cell).

The experimental design and overall workflow closely resembles the previous publication by Li et al., 2019. However, it comparatively lacks sufficient mechanistic insight for a publication in Nature Communications, but is perhaps suitable for one of the sister journals. The authors have previously

demonstrated that the CST complex protects stalled replication forks from aberrant MRE11-mediated nascent strand DNA degradation (NSD) (Lyu et al., 2021, EMBO). The authors have not provided sufficient further mechanistic insight into the function of STN1 phosphorylation to provide significant advances to our understanding of how the CST complex protects stalled replication forks. For example, the authors describe in their manuscript that S96 phosphorylation has no role in regulating the intrinsic DNA-binding activity of CST, and both the phosphor-inactive and -mimetic mutants still interact with RAD51 and are capable of inhibiting MRE11 degradation in vitro. The authors clearly state that there is a "to-be-identified" molecular function of the S96 phosphorylation, but the current manuscript lacks this mechanism. The manuscript could benefit from several added experiments to further strengthen their observations, and an addition of some experimental controls (especially for S96 antibody specificity validation). Finally, the level of support for the conclusions would be strengthened by repeating the key experiments over at least three biological replicates.

Major comments:

1. Where the authors report generating a custom S96 antibody for STN1 phosphorylation, the authors should add data either in the main figure or supplementary of further antibody validation. For example, as the antibody recognizes phosphorylated STN1 at S96, the authors are missing a Western Blot in the presence of phosphatase to show specificity. Also, the authors did not discuss the suitability of this antibody for immunofluorescence microscopy to perhaps validate the antibody in terms of its localization and demonstrate that phosphatase treatment gets rid of the signal (or changes in staining intensity in AMPK KO cells or in response to Calcium chelators). Furthermore, the authors show antibody specificity blot in WT and S96A mutants, but an additional good control would be a phosphor-mimetic mutant. Finally, please use at least two cell lines to validate the antibody.
3. Where the authors describe that STN1 is phosphorylated by both CaMKK2 and ATR-CHK1, my concern here is that there is no pathway specificity. It is unclear whether the two pathways compete with each other or are compensatory. Have the authors inhibited both the CaMKK2-dependent and the ATR-CHK1-dependent pathways together to see if there is any residual S96 signal upon replication stress? Further experiments are needed to determine this.
4. In Figure 1, the authors could perform an additional BrdU assay for single-stranded DNA quantifications between control and HU-treated conditions, with or without WT-STN1 and IDR mutant conditions. It would strengthen the results presented in Figure 1B.
5. In Figure 2F, it would be more informative to plot the graph with the number of chromosomal aberrations per condition (e.g. 0, 1-10, >10). This would give a better idea of the severity of aberrations between different conditions, and is a more quantitative way of assessing differences.
6. In Figure 4B, could the authors also add an additional condition showing that the induced S96 phosphorylation is removed upon phosphatase inhibitors? Furthermore, a similar agent to Thapsigargin, such as Tunicamycin, can be used to further strengthen their data.
7. In Figure 4, if the S96 antibody works for immunofluorescence microscopy, the authors could further strengthen their results by observing S96 staining upon Thapsigargin/Tunicamycin and seeing a rescue with a calcium chelator BAPTA.
8. In Figure 4B, the authors could include AMPK activation marker as one of the controls.
9. Can the authors perform a clonogenic survival assay in WT and AMPK KO cells to see if cell survival is rescued in the presence of WT STN1 versus phosphor-mutant in response to HU? It would provide additional functional relevance of this modification.
10. In Figure 5B, please repeat the experiment with either EdU staining or S-phase marker staining, and count the S-phase population to make sure that the same population of cells is quantified and variations from the cell cycle are minimized.
11. Similar papers in the field publishing the SIRF assay have used 10 micromolar EdU concentration (e.g. Panigua et al., 2022, Nat Commun). How are the authors justifying using such a high dose of EdU (125 micromolar)? The high dose used is a bit concerning in terms of the cellular effects it has.

Minor comments:

1. Please add additional reference(s) after the first sentence in the Introduction (PDF page 3)
2. Could the authors please make sure that where siRNA is used, the main phenotypes are validated with at least two siRNAs sequences.
3. Figure 2D is perhaps best for supplementary data as it is just to validate results shown in Figure 2C in another cell line.
4. In the Discussion section on page 14 of the PDF, the manuscript would benefit from a more nuanced discussion comparing the CaMKK2-mediated EXO1 activation to the current observations with STN1 S96. For example, how these two phosphorylation events might contribute in different ways to replication fork protection.
5. The authors should repeat all the key experiments at least over three biological replicates.
6. The HU concentrations are changing between some experiments (4 mM versus 10 mM). It is important to include only consistent data with one selected HU concentration (preferably 4 mM).
7. The DAPI channel and immunofluorescence images in general are overexposed and completely saturated. Especially when quantifying data in Figure 4A, it is important to have non-saturated signal for accurate comparisons. Same applies for SIRF immunofluorescence data.
8. As the authors have many experiments using the SIRF assay, can the authors have a small outline of how the SIRF assay works on top of one of the figures?
9. Regarding SIRF immunofluorescence, please also show the red channel separately in addition to merge with DAPI.
10. In Figure 5B where RAD51 foci are shown, please indicate all treatments / treatment doses and recovery times on the Figure.
11. In the final working model in Figure 7, please make the figure more clear: please draw out the CST complex in a more visible way (making it bigger and labeling both the proteins and the S96 phosphorylation mark). Also, perhaps adding a box outlining the fork protective functions of S96 phosphorylation (e.g. in terms of extensive ssDNA, chromosomal instability, etc). Finally, please indicate also AMPK in this model as it is used several times in the manuscript.

Point-by-point response:

Dear Reviewers,

We are grateful to you for your insightful and constructive comments that have helped us strengthen our manuscript. As detailed below, we have addressed all the issues raised by the reviewers. Our responses are in blue below. We also marked major changes in the main text in the manuscript in red. We hope you will find the revised version much improved.

Reviewer #1 (Remarks to the Author):

In this study, Jaiswal et al. reveal how the phosphorylation on STN1-S96 regulates the function of the CST complex at replication forks upon replication stress. Phosphorylation of STN1 is induced by replication stress and mediates STN1 fork localization to prevent nascent strand DNA degradation by MRE11. STN1 phosphorylation also promotes RAD51 localization to forks for genome stability. Combining cell-based and in vitro biochemical analysis, the authors show that ATR-CHK1 and CaMKK2 are potential kinases for the phosphorylation. Cancer mutations around the phosphorylation site phenocopy the effect of the phosphor-inactive mutation, implicating a role of STN1 in cancer development.

Overall, the authors provide evidence that S96P-dependent STN1 fork localization is crucial for CST function to protect stalled replication forks and promote genome stability. However, the mechanism of how phosphorylation controls STN1 localization was not characterized. Recent structural studies have provided functional insights into how CST complex functions to genome maintenance (PMID: 35830881; PMID: 35578024). It is shown that the N-terminus of STN1 not only interacts with CTC1 and TEN1 for CST complex formation but is also intensively involved in the contacts with the polymerase alpha-primase and DNA template to regulate de novo DNA synthesis. If the authors can investigate whether the role of S96 phosphorylation is linked to polymerase alpha-primase regulation, they may have a chance to provide better conceptual advances for the cellular function of the CST complex.

Thank you for this point. Please see our detailed response to your Point #2 below.

Major comments:

1. The authors show that ATR-CHK1 and CaMKK2 phosphorylate STN1 (Fig. 4). It is essential to know whether inhibition of ATR-CHK1 and CaMKK2 prevents STN1 localization at stalled forks and increases NSD.

We thank you for this very important point. We first checked the effect of single inhibition and co-inhibition of ATR-CHK1 with CaMKK2 on STN1 phosphorylation. Our results show that single inhibition of ATR-CHK1 or CaMKK2 reduced STN1 phosphorylation, and co-inhibition further reduced the phosphorylation, as shown in the new Figure 5A. We then conducted SIRF assays after separately inhibiting ATR, CHK1, and CaMKK2, as well as in various combinations. Our findings revealed that STN1 localization decreased after the inhibition of ATR, CHK1, and

CaMKK2 individually. Intriguingly, when we simultaneously blocked both the ATR-CHK1 and CaMKK2 pathways under the perturbed replication condition, STN1 localization at the stalled fork further decreased. These observations provide additional evidence that both pathways independently phosphorylate STN1.

2. It has been shown that nuclear localization of CST is regulated during the cell cycle and by polymerase alpha-primase interaction (PMID: 33731801). It is important to know if STN1 phosphorylation affects CST nuclear localization and polymerase alpha-primase interaction.

We appreciate this valuable suggestion. To investigate the potential impact of S96 phosphorylation on the interaction between Pol α and CST, we conducted co-IP experiments. Our co-IP results show that S96 phosphorylation does not seem to affect the interaction between Pol α and CST, as there were no significant changes observed in Pol α binding with WT-STN1, S96D, and S96A. This data is provided in the new Figure 6C.

Furthermore, we examined whether S96 phosphorylation has any influence on CST's nuclear localization. Unfortunately, the anti-pS96 antibody shows a strong non-specific recognition of an unknown protein in cell lysates (western blot is provided in Supplementary Figure S6), preventing us from using this antibody to detect pS96 localization using immunostaining. We thus performed nuclear and cytoplasmic fractionation of cells expressing WT-STN1, S96D, and S96A. Our results indicate that S96 phosphorylation seems to have little impact on the nuclear localization of STN1 (new Figure 6D).

3. In Fig. 7, the authors investigated the effect of cancer-associated missense mutations on STN1-S96 phosphorylation by western blots using the phosphor-S96 antibody. The data showed that the STN1-E95G mutant was not recognized by the antibody, so it was concluded that the E95G mutation prevents STN1 phosphorylation. However, an alternative explanation is that E95 is part of the epitope of the antibody and contributes to antibody binding. So E95G mutation can disrupt antibody binding without affecting S96 phosphorylation. To overcome the antibody issue, the authors can try to see if the antibody can recognize STN1-E95G/S96D.

Thank you for this suggestion. We generated the E95G/S96D mutant and assessed the specificity of the S96 antibody using Western blotting. The S96 antibody exhibited a substantial phosphorylation signal in E95G/S96D, while it was absent in the E95G mutant (new Figure 8B).

Minor comments:

1. Basal level of STN1 phosphorylation was detected in unperturbed cells and induced by HU, suggesting that STN1 phosphorylation is cell cycle regulated. Thus, it is necessary to check whether ATRi, CHKi, or CaMKK2i treatment affects cell cycle progression (Fig. 3 & 4). Because if kinase inhibitor pre-treatment reduces the S phase population, STN1 phosphorylation can be altered. So the authors need to exclude the possibility that the loss of STN1 phosphorylation is not just a secondary effect of kinase inhibition.

We appreciate your insightful inquiry. To address this, we conducted propidium iodide staining on cells subjected to treatment with ATR, CHK1, and CaMKK2 inhibitors both individually and in

various combinations. The results from the cell cycle analysis revealed that there were no significant alterations in the percentage of cells in the S-phase population. This data is provided in Supplementary Figure S5.

2. The paragraph title “ATR/CHK1 phosphorylates STN1 S96 in response to perturbed DNA replication (Line 228)” seems to be an overstatement.

We have toned down this sentence.

3. To determine the specificity of S96P antibody, phosphatase treatment on HU sample is recommended, better than S96A sample.

Thanks for asking this important question. Phosphatase treatment was done not only in WT-STN1, but also in S96D under the perturbed replication condition in two cell lines HeLa and U2OS. The S96 phosphorylation disappeared after phosphatase treatment in both WT and S96D, further validating the specificity of the antibody in recognizing S96 phosphorylation. This data is provided in the new Figure 3C.

Reviewer #2 (Remarks to the Author):

In this study, Jaiswal et al report that STN1 is phosphorylated by CaMKK2 at its intrinsic disordered region S96. The S96 phosphorylation is physiologically relevant and responsible for its localization at stalled forks. This is an interesting and important finding demonstrated by nice DNA fiber and biochemical assays. Furthermore, the current finding also links to STN1 cancer-associated STN1 variants that confer fork protection defects. Overall, this study is well-designed and executed. There are a few minor concerns that need to be addressed in the current manuscript.

1. The author has proposed a working model, however, this is still remains unclear to me how ATR/CHK1 and CaMKK2 function genetically on STN1. Additionally, the physiological end point is important to present in this study, e.g. whether the STN1 mutation(s) are hypersensitive to HU. Do the STN1 mutations affect cell cycle profile.

Thank you for this very important point. We have performed cell cycle analysis in cells expressing RNAi-resistant S96A, S96D, and WT-STN1 with concurrent depletion of endogenous STN1. Our results show that they have little significant effect on cell cycle progression. This data is provided in Supplementary Figure S7. We have performed colony formation assay and found that S96A mutation was unable to rescue the HU hypersensitivity induced by STN1 depletion and the S96D mutant fully rescued the HU hypersensitivity. This new data is provided in Supplementary Figure S2.

Regarding the functional relationship between ATR/CHK1 and CaMKK2 in phosphorylating STN1, we have provided more data that are now included in the new Figure 5. We think our new data support that the two pathways independently phosphorylate STN1 and regulate CST

protection of stalled forks from unwanted NSD. Please see our response to Reviewer #1 Major Point #1.

2. How is CaMKK2 activated under replication stress? is there any marker/evidence to show CaMKK2 is activated under the experimental conditions?

CaMKK2 activation occurs via the c-GAS-STING pathway, as previously documented by our co-author (PMID: 36696898 PMID: PMC9974760). They have described how replication stress generates single-stranded (ssDNA) and double-stranded DNA (dsDNA) that, upon translocating into the cytoplasm, activates the sensor protein cGAS. This activation, in turn, initiates the synthesis of cGAMP. Consequently, the binding of cGAMP to STING results in the detachment of STING from TRPV2 transporter localized on the endoplasmic reticulum membrane. The liberated TRPV2 then undergoes activation, releasing calcium ions (Ca²⁺) from the endoplasmic reticulum. This Ca²⁺ release subsequently triggers a downstream signaling cascade, safeguarding against replicating fork degradation.

CaMKK2 activation can be detected by the activation of its downstream target AMPKalpha, which is phosphorylated by CaMKK2. Under our experimental condition, we detected increased pAMPKalpha after Thapsigargin treatment. Such activation was inhibited by the calcium chelator BAPTA-AM. This new data is provided in Supplementary Figure S3.

3. Figure 3E: the control for ATR protein level is missing

We have now included the ATR western blot in our revised manuscript in the new Figure 3F (original Figure 3E).

4. What's the explanation for the KU80 level for A23187 treated cells?

KU80 was used as loading control in the original manuscript, as it is an abundant protein and was used by others as loading control. We have stripped our membranes and used GAPDH as loading control, and replaced KU80 blots with GAPDH blots in the revised manuscript.

5. Loading controls were missing in multiple blots.

We have used KU80 as a loading control as used by others. However, we have replaced them with either GAPDH or beta-actin blots in the revised manuscript.

6. Substandard data quality for several of the western blots, which include Figures 1B, 4F, and 7A.

The S96 antibody gives a very strong nonspecific band at around 56 kDa. We now provided two full blots of S96 antibody in Supplementary Figure S6. Because the endogenous STN1 protein is in very low abundance, this strong nonspecific band causes high background, making it very challenging to detect pS96. We have replaced Fig 1B with a nice and clean blot. Considering this, we believe that the blot in Figure 4F is acceptable. Additionally, in Figure 7A, we have

successfully obtained cleaner blots for E95G, S96V, and WT-STN1, which enhances the overall quality of the data presentation.

Immunofluorescence images are also over-saturated and some of the scale bars are inconsistent and low resolution.

Corrections are done as per your suggestion.

7. There are typos across the manuscript.

We have corrected typos in our revised manuscript.

Reviewer #3 (Remarks to the Author):

The authors describe STN1 activation in the CTC1-STN1-TEN1 (CST) complex by the ATR-CHK1 and CaMKK2-AMPK signaling pathways in response to replication stress. They show that replication stress induces STN1 phosphorylation in its intrinsic region by both the ATR-CHK1 and the calcium-sensing kinase CamKK2, and an absence of STN1 activation leads to MRE11-mediated nascent strand degradation and overall genome instability. Although many ATR-CHK1 targets have been identified at stalled forks, so far EXO1 has been the only known target in the CaMKK2-AMPK signaling pathway (Li et al., 2019, Molecular Cell).

The experimental design and overall workflow closely resembles the previous publication by Li et al., 2019. However, it comparatively lacks sufficient mechanistic insight for a publication in Nature Communications, but is perhaps suitable for one of the sister journals. The authors have previously demonstrated that the CST complex protects stalled replication forks from aberrant MRE11-mediated nascent strand DNA degradation (NSD) (Lyu et al., 2021, EMBO). The authors have not provided sufficient further mechanistic insight into the function of STN1 phosphorylation to provide significant advances to our understanding of how the CST complex protects stalled replication forks. For example, the authors describe in their manuscript that S96 phosphorylation has no role in regulating the intrinsic DNA-binding activity of CST, and both the phosphor-inactive and -mimetic mutants still interact with RAD51 and are capable of inhibiting MRE11 degradation in vitro. The authors clearly state that there is a "to-be-identified" molecular function of the S96 phosphorylation, but the current manuscript lacks this mechanism. The manuscript could benefit from several added experiments to further strengthen their observations, and an addition of some experimental controls (especially for S96 antibody specificity validation). Finally, the level of support for the conclusions would be strengthened by repeating the key experiments over at least three biological replicates.

Major comments:

1. Where the authors report generating a custom S96 antibody for STN1 phosphorylation, the authors should add data either in the main figure or supplementary of further antibody

validation. For example, as the antibody recognizes phosphorylated STN1 at S96, the authors are missing a Western Blot in the presence of phosphatase to show specificity. Also, the authors did not discuss the suitability of this antibody for immunofluorescence microscopy to perhaps validate the antibody in terms of its localization and demonstrate that phosphatase treatment gets rid of the signal (or changes in staining intensity in AMPK KO cells or in response to Calcium chelators). Furthermore, the authors show antibody specificity blot in WT and S96A mutants, but an additional good control would be a phosphor-mimetic mutant. Finally, please use at least two cell lines to validate the antibody.

We thank the reviewer for their suggestion. We have now included data showing that phosphatase treatment removes the pS96 signal (new Figure 3C). Two cell lines HeLa and U2OS were used as this reviewer suggested (new Figure 3C). We have also included phosphor-mimetic S96D mutant and observed that the pS96 antibody recognizes S96D like WT (new Figure 3C).

Regarding the use of the S96 antibody for immunofluorescence, we encountered challenges due to its strong non-specific signal at approximately 56 kD (shown in Supplementary Figure S6). Consequently, this antibody is not suitable for immunofluorescence assays. We now state this in the revised manuscript.

We conducted Western blot where we utilized the calcium chelator (BAPTA-AM) and found a substantial reduction in the pS96 signal, along with a corresponding decrease in pAMPK α following BAPTA-AM treatment, as shown in Supplementary Figure S3.

3. Where the authors describe that STN1 is phosphorylated by both CaMKK2 and ATR-CHK1, my concern here is that there is no pathway specificity. It is unclear whether the two pathways compete with each other or are compensatory. Have the authors inhibited both the CaMKK2-dependent and the ATR-CHK1-dependent pathways together to see if there is any residual S96 signal upon replication stress? Further experiments are needed to determine this.

We acknowledge the reviewer's thoughtful concern. Upon inhibiting both pathways, we observed a more pronounced decrease in S96 phosphorylation compared to when inhibiting just one of them. Results, provided in the new Figure 5A, show that single inhibition of ATR-CHK1 or CaMKK2 reduced STN1 phosphorylation, and co-inhibition further reduced the phosphorylation. In addition, we also checked STN1 localization at stalled forks upon separate inhibition of CaMKK2 or ATR-CHK1, or co-inhibition of the two pathways. As shown in the new Figure 5B, STN1 localization decreased after the inhibition of ATR, CHK1, and CaMKK2 individually. Intriguingly, when we simultaneously blocked both the ATR-CHK1 and CaMKK2 pathways, STN1 localization at the stalled fork further decreased. These observations provide additional evidence that both pathways independently phosphorylate STN1.

4. In Figure 1, the authors could perform an additional BrdU assay for single-stranded DNA

quantifications between control and HU-treated conditions, with or without WT-STN1 and IDR mutant conditions. It would strengthen the results presented in Figure 1B.

We have included the BrdU staining data per your suggestion in Supplementary Figure S1.

5. In Figure 2F, it would be more informative to plot the graph with the number of chromosomal aberrations per condition (e.g. 0, 1-10, >10). This would give a better idea of the severity of aberrations between different conditions, and is a more quantitative way of assessing differences.

We appreciate your feedback. Unfortunately, the chromosome spread images were acquired using the automatic MetaSystems microscope at PI's previous institution. After PI moved to the current institution, we do not have access to the MetaSystems or the associated software to open these images. Nevertheless, we believe that our current graph is sufficient in supporting the major conclusion of the manuscript.

6. In Figure 4B, could the authors also add an additional condition showing that the induced S96 phosphorylation is removed upon phosphatase inhibitors? Furthermore, a similar agent to Thapsigargin, such as Tunicamycin, can be used to further strengthen their data.

We value your thoughtful input. We have included phosphatase data into our revised Figures 3C and 4B. In our initial manuscript submission, we employed two Ca²⁺ inducers, namely Ca²⁺ ionophore and thapsigargin, and observed consistent results. We believe that the inclusion of an additional drug would not yield significant advancement in the study.

7. In Figure 4, if the S96 antibody works for immunofluorescence microscopy, the authors could further strengthen their results by observing S96 staining upon Thapsigargin/Tunicamycin and seeing a rescue with a calcium chelator BAPTA.

We appreciate your thoughtful consideration. As previously explained, we consistently obtain a robust band at approximately 56 kD when utilizing the pS96 antibody. Consequently, this antibody couldn't be employed for immunofluorescence experiments. Please see our response to your Major Point # 1.

8. In Figure 4B, the authors could include AMPK activation marker as one of the controls.

We have provided the western blots of AMPK activation (as indicated by increased phosphor AMPK) by thapsigargin in Supplementary Figure S3.

9. Can the authors perform a clonogenic survival assay in WT and AMPK KO cells to see if cell survival is rescued in the presence of WT STN1 versus phosphor-mutant in response to HU? It would provide additional functional relevance of this modification.

We appreciate your thoughtful consideration. We do not entirely understand this suggestion, since our data show that AMPK KO does not reduce STN1 phosphorylation. We guess what this reviewer meant was to check if WT STN1 and phosphor-mutant could rescue **CaMKK2** KO cells

in cell survival. As noted in our manuscript, the CaMKK2 signaling pathway not only phosphorylates STN1 but also EXO1. Moreover, it is highly likely that the CaMKK2 pathway has additional targets that are yet to be identified. With these factors in mind, we believe that WT-STN1 alone will not be sufficient to rescue cell survival of CaMKK2 KO cells.

10. In Figure 5B, please repeat the experiment with either EdU staining or S-phase marker staining, and count the S-phase population to make sure that the same population of cells is quantified and variations from the cell cycle are minimized.

We have conducted EdU staining in the same set of cells, and found no noticeable changes in the proportion of cells in the S-phase. New data are provided in Supplementary Figure S5.

11. Similar papers in the field publishing the SIRF assay have used 10 micromolar EdU concentration (e.g. Panigua et al., 2022, Nat Commun). How are the authors justifying using such a high dose of EdU (125 micromolar)? The high dose used is a bit concerning in terms of the cellular effects it has.

Thanks for pointing this out. After conducting a literature review, we discovered that researchers have employed two distinct concentrations of EdU in their studies. Some have used 10 μ M, while others have opted for 125 μ M. Importantly, the original publication that developed SIRF technique used 125 μ M (<https://doi.org/10.1083/jcb.201709121>). Here we provide a few examples in which 125 μ M of EdU was utilized for you to check. <https://doi.org/10.1016/j.molcel.2020.08.018> <https://doi.org/10.1073/pnas.2121406119> <https://doi.org/10.1093/nar/gkac808>; <https://doi.org/10.7554/eLife.39140>.

Minor comments:

1. Please add additional reference(s) after the first sentence in the Introduction (PDF page 3)

We have included the additional references in our revised manuscript.

2. Could the authors please make sure that where siRNA is used, the main phenotypes are validated with at least two siRNAs sequences.

In our study, we performed the rescue experiment using RNAi-resistant STN1 to show that the phenotypes caused by siRNA can be rescued by WT. This approach is widely recognized as the best way to validate the specificity of siRNA. With due respect, we do not think using a second siRNA is necessary.

3. Figure 2D is perhaps best for supplementary data as it is just to validate results shown in Figure 2C in another cell line.

Thanks for your suggestion. BJ/hTERT is a non-cancer cell line. We feel that it is important to show that non-cancer cell lines also show the same results as cancer cell lines. So we leave Figure 2D in the main figure.

4. In the Discussion section on page 14 of the PDF, the manuscript would benefit from a more nuanced discussion comparing the CaMKK2-mediated EXO1 activation to the current observations with STN1 S96. For example, how these two phosphorylation events might contribute in different ways to replication fork protection.

We value your thoughtful input. We have discussed this in our revised manuscript.

5. The authors should repeat all the key experiments at least over three biological replicates.

We appreciate your careful consideration. It's worth noting that all the critical experiments, including but not limited to DNA fiber analysis, the majority of SIRF experiments, in vitro kinase assay, HU/calcium ionophore/thapsigargin treated pS96 western, western blots in CaMKK2 KO and AMPK KO cells, cancer mutations, co-IP, in vitro DNA binding assays, in vitro MRE11 degradation assays, metaphase spread, and RAD51 immunostaining, have been repeated three times independently to ensure reproducibility. We have indicated this information in corresponding figure legends and the "Western blotting and antibodies" section in Materials and Methods.

In addition, for critical DNA fiber and SIRF experiments, the images were analyzed independently by two individuals in the lab to avoid human bias. We have indicated this information in the "DNA fiber assay" and "SIRF assay" in Materials and Method.

6. The HU concentrations are changing between some experiments (4 mM versus 10 mM). It is important to include only consistent data with one selected HU concentration (preferably 4 mM).

In our initial experiments, 10 mM HU was used. Later on when we used 4 mM HU, we observed similar pS96 stimulation and we then switched to 4 mM treatment. In the majority of results presented in our revised manuscript, 4 mM HU were used. We have indicated HU treatment condition in Figure legends.

7. The DAPI channel and immunofluorescence images in general are overexposed and completely saturated. Especially when quantifying data in Figure 4A, it is important to have non-saturated signal for accurate comparisons. Same applies for SIRF immunofluorescence data.

Changed.

8. As the authors have many experiments using the SIRF assay, can the authors have a small outline of how the SIRF assay works on top of one of the figures?

We value your significant suggestion. To address this, we have incorporated a figure outlining the SIRF assay in our revised manuscript in the revised Figure 1C.

9. Regarding SIRF immunofluorescence, please also show the red channel separately in addition to merge with DAPI.

In our revised manuscript, we have displayed the red channel separately as per your suggestion in Supplementary Figures.

10. In Figure 5B where RAD51 foci are shown, please indicate all treatments / treatment doses and recovery times on the Figure.

We thank the reviewer for this question. We have included this information in figure 5B legend.

11. In the final working model in Figure 7, please make the figure more clear: please draw out the CST complex in a more visible way (making it bigger and labeling both the proteins and the S96 phosphorylation mark). Also, perhaps adding a box outlining the fork protective functions of S96 phosphorylation (e.g. in terms of extensive ssDNA, chromosomal instability, etc). Finally, please indicate also AMPK in this model as it is used several times in the manuscript.

We value your significant suggestion. We have modified Figure 7 (revised Figure 8) to make CST more visible as per your suggestion.

Regarding the suggestion of including AMPK, we think it is better to leave it out, since we have shown that STN1 phosphorylation is independent of AMPKalpha. The only known target of AMPKalpha is EXO1. If we included AMPKa in our model, we would need to include EXO1, which is not the focus of this manuscript.

REVIEWERS' COMMENTS

Reviewer #1 (Remarks to the Author):

Thanks to the authors for providing new data to answer my questions. I am satisfied with the improvement in the quality of the manuscript and excited about the insights into the molecular mechanism of CST complex in DNA replication.

Reviewer #2 (Remarks to the Author):

The revised manuscript has addressed most of my concerns and is significantly improved in the current version.

There are a few minor things, perhaps careless mistakes, need to be addressed and clarified

1. Inconsistent label for the ladders, Figure 2C ends STN1 size is 43 (Left) and significantly lower than 43 (right)
2. Figure 4F WT, seemingly there is no induction for pS96 after HU treatment.
3. Missing size label for Figure 5A
4. The ATR knockdown is relatively inefficient

Reviewer #3 (Remarks to the Author):

The revised manuscript addresses several raised questions and the improvements made are enough to significantly improve the manuscript. Where suggested experiments were not carried out, the authors provided sufficient reason for their decision.

Point-to-point response:

Again, we would like to express our gratitude to the reviewers for their valuable feedbacks and experimental suggestions.

Reviewer #1 (Remarks to the Author):

Thanks to the authors for providing new data to answer my questions. I am satisfied with the improvement in the quality of the manuscript and excited about the insights into the molecular mechanism of CST complex in DNA replication.

Response: Your input has greatly enhanced the quality of our manuscript.

Reviewer #2 (Remarks to the Author):

The revised manuscript has addressed most of my concerns and is significantly improved in the current version. There are a few minor things, perhaps careless mistakes, need to be addressed and clarified.

1. Inconsistent label for the ladders, Figure 2C ends STN1 size is 43 (Left) and significantly lower than 43 (right)

Response: We have fixed this in our revised manuscript.

2. Figure 4F WT, seemingly there is no induction for pS96 after HU treatment.

Response: We concur with the reviewer's observation. It's important to note that the response to HU treatment and the associated phosphorylation events can exhibit variability between different experimental runs. Nevertheless, it's worth highlighting that we have consistently detected a significant increase in pS96 phosphorylation after HU treatment in multiple other blots. This collective evidence strongly suggests that HU indeed induces pS96 phosphorylation.

3. Missing size label for Figure 5A

Response: Fixed.

4. The ATR knockdown is relatively inefficient.

Response: ATR is an essential protein for cell survival and DNA repair. Cells rely on ATR to detect and repair DNA damage, and complete loss of ATR can be lethal to cells. This could be the reason we observed modest ATR knockdown as cells have evolved multiple mechanisms to maintain ATR expression and activity, making it difficult to completely knock down ATR.

Reviewer #3 (Remarks to the Author):

The revised manuscript addresses several raised questions and the improvements made are enough to significantly improve the manuscript. Where suggested experiments were not carried out, the authors provided sufficient reason for their decision.

Thank you. Your suggestions improved our manuscript significantly.